# Web-Scale Visual Entity Recognition:
# An LLM-Driven Data Approach

**Mathilde Caron**   **Alireza Fathi**   **Cordelia Schmid**   **Ahmet Iscen**
Google DeepMind

## Abstract

Web-scale visual entity recognition, the task of associating images with their corresponding entities within vast knowledge bases like Wikipedia, presents significant challenges due to the lack of clean, large-scale training data. In this paper, we propose a novel methodology to curate such a dataset, leveraging a multimodal large language model (LLM) for label verification, metadata generation, and rationale explanation. Instead of relying on the multimodal LLM to directly annotate data, which we found to be suboptimal, we prompt it to reason about potential candidate entity labels by accessing additional contextually relevant information (such as Wikipedia), resulting in more accurate annotations. We further use the multimodal LLM to enrich the dataset by generating question-answer pairs and a grounded fine-grained textual description (referred to as "rationale") that explains the connection between images and their assigned entities. Experiments demonstrate that models trained on this automatically curated data achieve state-of-the-art performance on web-scale visual entity recognition tasks (*e.g.* +6.9% improvement in OVEN entity task), underscoring the importance of high-quality training data in this domain.

## 1   Introduction

Entities are at the core of how we represent and organize knowledge, as seen in prominent encyclopedias like Wikipedia, where each article is dedicated to a specific entity. In the field of computer vision, the task of visual entity recognition aims to identify entities within query images. This capability is not only a fundamental building block for various entity-aware visual understanding tasks, including "info-seeking" Visual Question Answering (VQA) [10, 22, 31] and News Content Understanding [4, 14, 24, 54], but also has numerous commercial applications. Despite the progress made in recent years, current models often struggle with web-scale visual entity recognition. These models, typically trained on free-form image captions [9, 15, 56], tend to hallucinate entities or output overly generic ones, leading to suboptimal performance. We hypothesize that the root of this issue lies in the lack of clean, large-scale training data specifically designed for visual entity recognition.

Recent efforts have been made to address this problem by transforming existing captioning datasets into entity recognition datasets [7, 25]. For example, Caron et al. [7] propose to match each Wikipedia entity name to most similar captions in a large image-caption database [9, 46] and use the corresponding images as visual examples for the considered entity. Although this method has resulted in state-of-the-art performance, it still has significant limitations: the resulting datasets are often noisy, with poor matching between the image content and the candidate entity. First, a source of mistake is due to the ambiguity of language. For example the entity *bishop of llandaff* may refer both to a person and to a flower species [18, 35]. Second, there is some noise inherent to the used image-caption dataset that affects the results. For example in Fig. 1(a), an image of a building is incorrectly linked to the Wikipedia entity *Negative equity*. This mismatch occurs because of the caption "*Worst areas for negative equity*", which is likely to have been extracted from an investment-related website. The irrelevant association of the caption with the building image results in the inaccurate connection to the Wikipedia entity. Third, the text embedding matching between

38th Conference on Neural Information Processing Systems (NeurIPS 2024).

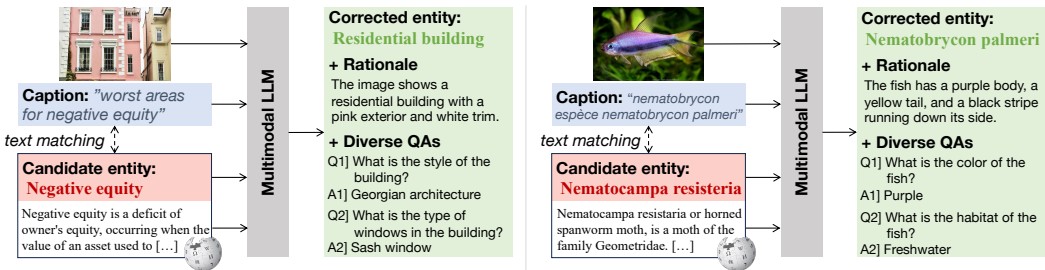

(a) Irrelevant caption      (b) Incorrect caption-entity text matching

Figure 1: **Two failure cases of the visual entity recognition dataset of [7].** Our proposed method overcomes these limitations by prompting a multimodal LLM to correct candidate entities. The LLM has access to relevant context such as the candidate entity Wikipedia page and the input image-caption pair. We also enrich the dataset with rationales and question/answer pairs covering diverse entities.

entity name and caption is not always accurate. For instance, in Fig. 1(b), the caption "*nematobrycon espèce nematobrycon palmeri*" is incorrectly matched with the entity name *nematocampa resistaria*, which are two distinct animal species (a fish and a moth). Moreover, another limitation of these datasets is that they typically focus on a single entity per image, which is a restrictive scenario, as most images contain multiple entities.

In this paper, we overcome these different limitations by proposing a novel methodology to curate a high-quality, large-scale dataset for web-scale visual entity recognition leveraging the capabilities of modern multimodal Large Language Models (LLMs) available through public APIs [2, 15, 37]. Our approach is unique in that we do not rely on the multimodal LLM for direct annotation, which we found to be suboptimal. Instead, we prompt the LLM to reason about candidate entity labels by accessing additional contextual relevant information, such as the original image captions and external knowledge sources like Wikipedia. This approach significantly improves the annotation quality of the resulting dataset, as evidenced by our thorough ablation studies. Moreover, we employ the multimodal LLM to augment our dataset with rationales that explain the relationship between images and their corresponding entities. We observe in our experiments that training on this additional metadata improves the performance and visual entity understanding of the models. Finally, we address the previous limitation of focusing on a single entity per image by prompting the multimodal LLM to generate several question-answer pairs that cover a diverse range of entities in the image.

We conduct extensive experiments to evaluate the effectiveness of our approach. The results demonstrate that models trained on our automatically curated data achieve state-of-the-art performance on web-scale visual entity recognition tasks, notably on the challenging Open-domain Visual Entity recognitioN (OVEN) benchmark [17] (*e.g.* +6.9% on the OVEN entity split and +3.8% on the OVEN query split). Remarkably, we obtain these results with moderate-size models, which are orders of magnitude smaller than competing approaches, highlighting the importance of high-quality training data in this domain. We further demonstrate the validity and effectiveness of our dataset when utilized as a memory base for approaches such as visual matching, showcasing its versability and potential for various applications. In summary, our contributions are threefold:

- We introduce a novel methodology to curate a large-scale dataset for web-scale visual entity recognition, using a multimodal LLM as a verification and annotation tool.
- We enrich the dataset with additional metadata, including question-answer pairs and rationales, generated by the multimodal LLM.
- We demonstrate the effectiveness of our approach with thorough ablation study and by achieving state-of-the-art performance on web-scale visual entity recognition tasks.

## 2 Related work

**Visual entity recognition.** Entities are key to knowledge representation and organization, as seen in prominent encyclopedias like Wikipedia. Visual entity recognition is the task of identifying entities based on visual queries [13, 43, 44], and is a critical component of many complex applications. One such application is info-seeking VQA, which involves providing answers to questions about the detailed properties of finegrained entities [10, 22, 31]. Another application is entity-aware captioning [28, 34, 61], a technique frequently employed in tasks such as news content comprehen-

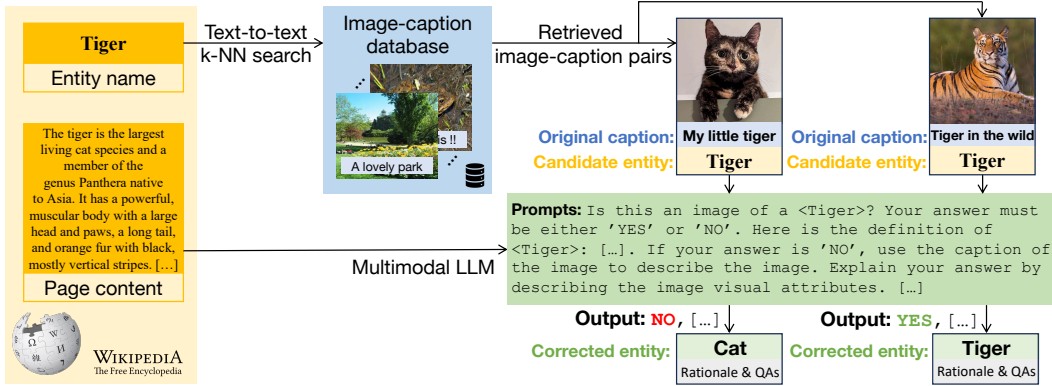

Figure 2: **LLM-Refined Entity-WebLI" (REW) dataset.** We propose a method to refine the Entity-WebLI dataset of Caron et al. [7] by prompting a multimodal LLM to verify and correct Wikipedia entities. We also prompt the multimodal LLM to output visually grounded rationales and question/answer pairs about diverse attributes of the image. Complete prompts are in Appendix A.3.1.

sion [1, 4, 14, 24, 54]. Recent research has expanded the scope of visual entity recognition to include web-scale and open-domain entities [17, 21, 41]. Of particular interest, Hu et al. [17] introduce the Open-domain Visual Entity recognitioN (OVEN) benchmark, which includes over 6 million entities from English Wikipedia, covering a broad range of concepts such as animals, buildings, organizations, landmarks, etc. Caron et al. [7] recently achieved state-of-the-art results on the OVEN benchmark by re-purposing auto-regressive generative models for this task. A key component of their approach is pretraining on a new entity-based dataset instead of captions. In this paper, we build on the work of [7] by improving their pretraining dataset using external multimodal LLMs.

**Using LLMs as annotation tools.** Since their remarkable success and widespread accessibility, LLMs [36, 40, 53] have been used in many different ways to obtain better supervision for training various tasks [23, 32, 45, 59, 60, 62]. For example, LLMs have been used to generate question-answer pairs from Wikipedia pages [8, 31] or from transcribed video narrations [59], while other works use LLMs to rephrase questions into sentences [60]. Of particular interest, Hsieh et al. [16] show that prompting LLMs to output rationales as additional supervision for training small models in a multi-task framework is an effective strategy. A recent related trend consists in prompting LLMs to generate high-quality instruction-following training samples [11, 50], an approach which has succesfully been extended to multimodal tasks [26]. Unlike our work, these approaches prompt LLMs with text input only while we feed both image and text to multimodal LLMs. Another key difference is that we use the multimodal LLM as a verification and correction tool based on candidate annotations rather than using its raw output as supervision, which we find to be suboptimal for the task of web-scale finegrained entity recognition.

## 3 Method

In this section, we describe how to leverage the capabilities of modern multimodal LLMs [15] in conjunction with external knowledge repositories such as Wikipedia to create a clean, large-scale training dataset specifically designed for web-scale visual entity recognition. We refer to the resulting dataset as "LLM-Refined Entity-WebLI" (REW) and an overview of our method is in Fig. 2.

### 3.1 Preliminaries

**Wikipedia-scale visual entity recognition.** Following recent research in web-scale visual entity recognition [7, 17], our goal is to train models capable of accurately matching any given image-text query $(x_v, x_t)$ to an entity $e$ among a vast finegrained set $\mathcal{E}$ of possible entities. In this work, unless specified otherwise, the set of entities $\mathcal{E}$ consists of the 6 million entities from English Wikipedia. Each entity $e \in \mathcal{E}$ comes with an entity name $t_e$ corresponding to the entity Wikipedia page title.

**Entity-based pretraining dataset.** Previous works have observed that auto-regressive image captioning models like GIT [56] or PaLI [9] have suboptimal results when transferred to visual entity recognition (see Tab. 1), due to the differences between captioning and entity recognition tasks [7, 17]. To address this, Caron et al. [7] propose training such models on entity-based data, not captions, and automatically create a new dataset called "Entity-WebLI" for this purpose. In short, for each

Wikipedia entity, the authors find the most relevant image-caption pairs in WebLI through nearest neighbor search in the CLIP text embedding space [41] between encoded entity names and captions. They then replace the captions of the corresponding retrieved images with the considered entity name. In this work, we refer to the entities obtained in this manner as "candidate entities". Further details on the Entity-WebLI dataset are in [7]. We describe in the following how to improve over such a dataset by leveraging the capabilities of modern multimodal LLMs.

## 3.2 Generating finegrained entities and descriptions with multimodal LLMs

**Entity verification and correction.** Our goal is to overcome the limitations of existing entity-based pretraining dataset discussed in the introduction. To improve the correspondence quality between images and entities, we propose to use existing multimodal LLMs to verify the assignment of an entity to an image. In particular, given an image $x_v$ and corresponding entity $e$ from the Entity-WebLI dataset, we prompt a multimodal LLM with the task of verifying if $x_v$ is an image representing entity $e$. Interestingly, we make two important findings in our experiments (see Tab. 4). First, directly predicting an entity $e$ is a more challenging problem than verifying the validity of the proposed entity. Intuitively, this is because multimodal LLMs either output too generic entities or hallucinate when they do not know the correct entity. This is expected since LLMs haven't specifically been trained to identify entities at very high levels of granularity. This results in sub-optimal performance for the finegrained entity recognition tasks, where the goal is to recognize non-generic finegrained entities.

Second, we find that the verification by a multimodal LLM is more precise when it has access to external metadata such as the page content of the candidate Wikipedia entity $e$ and the original caption. Intuitively, the Wikipedia content allows the multimodal LLM to know which visual attributes to look for in the image $x_v$, while the original caption gives hints about the corrected entity when the multimodal LLM detects that the candidate $e$ is not correct. Note that the corrected entity might not always coincide with an actual Wikipedia entity since we do not provide the list of the 6M Wikipedia entities to the LLM. We still include these corrected entities in our training dataset and simply constrain decoding to the 6M entities at inference time (see details in Sec. 4.1). Similar techniques have also been used into entity generation in multimodal contexts by works like GMEL [48] and AutoVER [58]. Our prompt for entity verification and correction is available in Appendix A.3.1.

**Generating rationales.** At the same time as asking the multimodal LLMs to verify the entities assigned to training images, we also prompt it to provide a visually grounded rationale for its proposed entity. This allows to explain the connection between the image and the assigned entity as can be seen in the examples in Fig. 3 and Fig. 4. We observe in our experiments that it improves the performance of the models for entity recognition (see Tab. 4). When training our model (see details in Sec. 3.3), we follow a multi-task learning strategy where we prepend task prefixes to the input examples so that the model can output differently based on whether it is asked to predict an entity name or a rationale.

**Generating question-answer pairs (QAs).** The Wikipedia entity recognition OVEN benchmark [17] consists of two splits: an entity split and a query split. In the query split, images typically contains multiple entities, and the input question $x_t$ determines which entity should be recognized by the visual entity recognition models. We observe in our early experiments and in Tab. 1 that training on the Entity-WebLI dataset [7] consistently results in poor performance in the query split, while achieving state-of-the-art performance for the OVEN entity split. We hypothesize that this is due to how Entity-WebLI dataset is constructed (see details in Sec. 3.1). The k-NN search in the text embedding space favors short captions containing entity names. As a result, a single, unambiguous, entity is assigned to each image, a scenario quite different to the examples in the query split.

To overcome this problem, we prompt the multimodal LLM to generate question-answer pairs for several, diverse entities in the input image (see prompt in Appendix A.3.1). The multimodal LLM has access to the input image, the verified/corrected entity as well as the rationale it previously generated. We empirically evaluate in the ablation study in Tab. 5 the impact of these on the question/answer pairs and resulting trained models.

## 3.3 Model training

**Auto-regressive generative models.** Previous works have shown the effectiveness of *generative* approaches for entity recognition both in the pure NLP domain [12, 30, 39, 42, 49, 51] and, more recently, in visual entity recognition benchmarks [7, 17]. Motivated by their success, we perform entity recognition by generating Wikipedia entity names (*i.e.* page titles $t_e$) in an auto-regressive

fashion. This is akin to a multimodal version of GENRE [12] or to the GER-CAPTION variant of [7]. Formally, we transform an input image-text query pair $x = (x_v, x_t)$ into a set of $N$ $d$-dimensional embeddings $\mathbf{X} \in \mathbb{R}^{N \times d}$ formed by concatenating the visual encoder output of $x_v$ and the text tokenizer output of $x_t$. We use an auto-regressive text decoder $g(\cdot)$ to generate the target text $y$. As detailed in the following paragraph, the target text can either be an entity name or a rationale. Specifically, the decoder predicts each text token $y_k$ from the target text (total length is $K$) given both the set of preceding token embeddings $\mathbf{Y}_{<k}$ and the input image-text embeddings $\mathbf{X}$. We train with a language modeling objective:

$$\mathcal{L} = \frac{1}{K} \sum_{i=1}^{K} \ell(y_k, g([\mathbf{X}; \mathbf{Y}_{<k}])) \tag{1}$$

where $[;]$ corresponds to the concatenation operation in the first dimension and $\ell$ is the softmax cross-entropy loss with label-smoothing [33]. We average this loss over minibatches of examples and update the weights of the visual encoder and text decoder with back-propagation.

**Multi-task learning.** We train our models jointly with three distinct tasks: (i) predicting the verified/corrected visual entity, (ii) generating the rationale and (iii) answering the questions generated by the multimodal LLM. These three tasks are text generation tasks and follow the same framework introduced in the previous paragraph. They differ in the nature of the input text $x_t$ and target text $y$. For entity recognition and question answering, the target text $y$ corresponds to a possible entity name and the input text $x_t$ corresponds to a question. For rationale generation, the target text $y$ corresponds to the rationale generated by multimodal LLM and the input text $x_t$ consists simply of a prefix specifying to the model that this task is distinct from entity generation. Our final multi-task loss objective is:

$$\mathcal{L}_{\text{Final}} = \mathcal{L}_{\text{Entity}} + \mathcal{L}_{\text{Rationale}} + \mathcal{L}_{\text{QA}}$$

where $\mathcal{L}_{\text{Entity}}$, $\mathcal{L}_{\text{Rationale}}$ and $\mathcal{L}_{\text{QA}}$ correspond to entity, rationale and answer generation respectively.

## 4 Experiments

### 4.1 Experimental setting

We detail here the most salient experimental details. Full experimental setting is in Appendix A.3.

**REW training dataset.** Our training dataset builds upon the Entity-WebLI dataset [7] (see Sec. 3.1) which itself is based on WebLI [9], a dataset already deduplicated against the train, val, and test splits of 68 common vision/vision-language datasets [9]. The Entity-WebLI dataset is further aggressively filtered against OVEN by removing any image which has a CLIP-score higher than 0.95 with an OVEN image [7]. This ensures that there is no downstream data leakage in our REW dataset. We build two versions of REW dataset: REW-5M (4.5M images) and REW-47M (47M images). Each training image is attached to a verified/corrected entity, a rationale and 3 question/answer pairs. In the main results section (Sec. 4.2) we train on REW-47M dataset for $600k$ steps while for analysis and ablation study (Sec. 4.3) we train on REW-5M for a shorter schedule ($200k$ steps). We also validate our dataset refining methodology using the LAION dataset [46] as the image-caption base dataset.

**Downstream task: OVEN.** We consider the entity and query splits of the OVEN benchmark [17]. For both splits, the goal is to output a Wikipedia entity given an input image and a question. OVEN validation and test splits are divided into seen and unseen entities. The seen examples correspond to entities present in the OVEN training set while unseen entities are a subset of entities not present in the training set. We report the harmonic mean (HM) of top-1 accuracy scores between "seen" and "unseen" entities [17]. We specify in our results if we report results before or after additional finetuning on the training set of OVEN ("+ seen finetune").

**Downstream task: finegrained datasets.** We also report results on Oxford Flowers [35], Sun397 [57], Food101 [5], FGVC-Aircraft [29] and Sports100 [17] finegrained datasets in the zero-shot mode. We choose these datasets since there is a direct mapping between their class vocabularies and the Wikipedia entities that our model is trained to output. The resulting class label to Wikipedia entity mappings are given in Appendix A.5.

**Inference.** We perform decoding with beam search. When evaluating on OVEN, we discard all decoded texts that are not one of the 6M Wikipedia entities. This is akin to constraining the beam search only at the last decoding step. Constraining from the first decoding step is too costly in our

Table 1: **Comparison with the state of the art on OVEN entity and query test splits.** We report the harmonic mean (HM) of the seen and unseen sets (top-1 accuracy) before and after finetuning on OVEN training seen categories ("+ seen finetune"). We indicate model architectures and their total number of parameters ("# par.") in billions as well as the training dataset details.

| Model | #par (B) | Dataset | Entity split | | | + seen finetune | | | Query split | | | + seen finetune | | |
|---|---|---|---|---|---|---|---|---|---|---|---|---|---|---|
| | | | HM | seen | unseen | HM | seen | unseen | HM | seen | unseen | HM | seen | unseen |
| *Dual encoders* | | | | | | | | | | | | | | |
| CLIPfusion [17] | 0.9 | OpenAI [41] | 5.2 | 5.6 | 4.9 | 8.4 | 33.6 | 4.8 | 1.6 | 1.3 | 2.0 | 2.7 | 25.8 | 1.4 |
| CLIP2CLIP [17] | 0.9 | OpenAI [41] | 5.2 | 5.6 | 4.9 | 11.5 | 12.6 | 10.5 | 1.6 | 1.3 | 2.0 | 3.5 | 3.8 | 3.2 |
| *Generative approaches* | | | | | | | | | | | | | | |
| PaLI-3B [9] | 3 | WebLI-1B [9] | – | – | – | 9.1 | 19.1 | 6.0 | – | – | – | 16.7 | 27.4 | 12.0 |
| PaLI-17B [9] | 17 | WebLI-1B [9] | 1.8 | 3.3 | 1.2 | 16.0 | 28.3 | 11.2 | 9.2 | 14.1 | 6.8 | 27.1 | 36.2 | 21.7 |
| GiT-Large [56] | 0.4 | WebLI-100M [9] | 2.1 | 4.7 | 1.4 | 6.5 | 13.7 | 4.2 | 3.9 | 5.1 | 3.2 | 15.6 | 28.9 | 10.7 |
| GER-ALD [7] | 0.4 | Entity-WebLI [7] | 17.7 | 18.3 | 17.2 | 22.7 | 31.5 | 17.7 | 6.3 | 6.0 | 6.7 | 5.8 | 14.1 | 3.6 |
| GiT-Large [56] | 0.4 | Entity-WebLI [7] | 19.1 | 19.8 | 18.5 | 20.1 | 25.9 | 16.4 | 10.4 | 9.8 | 11.0 | 10.1 | 17.7 | 7.1 |
| GiT-Large [56] | 0.4 | REW-47M (Ours) | **23.6** | **25.7** | **21.7** | **29.6** | **36.0** | **25.1** | **30.0** | **31.2** | **28.9** | **30.9** | **39.2** | **25.5** |

Table 2: **Zero-shot transfer of generative models to finegrained image classification.** We report top-1 accuracies. All models are run by us and are based on the same architecture.

| Model | Training dataset | Flowers [35] | Sun397 [57] | Food [5] | Aircraft [29] | Sports100 [17] |
|---|---|---|---|---|---|---|
| GiT-Large [56] | WebLI-100M [9] | 39.1 | 45.8 | 55.7 | 7.4 | 57.9 |
| GiT-Large [56] | Entity-WebLI [7] | 79.8 | 45.1 | 66.5 | 27.7 | 77.2 |
| GER-ALD [7] | Entity-WebLI [7] | 86.7 | 45.9 | 78.0 | 37.4 | 74.6 |
| GiT-Large [56] | REW-47M (Ours) | **88.2** | **50.2** | **80.4** | **50.3** | **78.0** |

implementation with such a large million-scale label space. For finegrained datasets, we perform constrained beam search decoding at all decoding steps since the label spaces are smaller.

**Model training implementation details.** We use GiT-Large [56]: it consists of a visual encoder (CLIP-L/14 [41]) and a 6-layer text decoder with internal dimension $d = 768$. Following [7], the visual encoder is first pre-trained jointly on WebLI-100M [9] and Conceptual Captions-12M [47] while the decoder is randomly initialized. We use batch size of 4096, learning rate of $1e^{-5}$ for the visual encoder and $1e^{-4}$ for the decoder, label smoothing of $0.2$ and no weight decay. We use standard inception crop data augmentation. For the multimodal LLM, we use Gemini Pro [15]. The public API is available at ai.google.dev.

## 4.2 Main results

**Comparison with the state of the art on OVEN.** In Tab. 1, we observe that PaLI and GiT-Large models pretrained on captioning datasets have suboptimal performance, especially in the entity split. Intuitively this is because the entity split tackles finegrained entity recognition while query split is more reflective of a generic VQA task, which leverages the language understanding abilities learned from captioning. Hence, the query split task is more aligned with the captioning pretraining than the entity split task. In Tab. 1, we see that the model trained on our REW dataset instead of captioning data results in state-of-the-art performance in the OVEN benchmark, both before and after further OVEN finetuning on the "seen" classes. Notably, our model outperforms the captioning PaLI-17B model by large margins: +13.6 top1 HM test accuracy on the entity split and +3.8 on the query split, while using $42\times$ less parameters.

Finally, we report the zero-shot performance of the multimodal LLM on OVEN: it reaches 13.3 HM top-1 in the entity split and 29.5 HM top-1 in the query split. These numbers suggest that we are not merely distilling from the considered multimodal LLM as we outperform its performance on this benchmark by +10.3 on entity and +1.4 on query sets while using orders of magnitude less parameters. The analysis in Tab. 4 will further demonstrate the importance of using the multimodal

Table 3: **Visual matching.** We report top-1 accuracies of visual matching with CLIP or DINOv2 ViT-L/14 visual backbones. We compare two types of annotations for the visual matching memory database: either the candidate entities or our multimodal LLM corrected entities. We report the *absolute* improvements of using the latter compared to the former between parentheses as well as the *relative* improvement averaged across the six datasets in the last column.

| Memory dataset | OVEN-Ent | Flowers | Sun397 | Food101 | Aircraft | Sports100 | Avg.relative Δ |
|---|---|---|---|---|---|---|---|
| *CLIP-L/14 backbone [41]* | | | | | | | |
| Candidate entities | 16.3 | 81.1 | 37.7 | 80.9 | 42.1 | 70.4 | – |
| + multimodal LLM correction | 19.8(+3.5) | 81.1(+0.0) | 49.7(+12.0) | 79.1(-1.8) | 44.6(+2.5) | 74.8(+4.4) | +10.5% |
| *DINOv2-L/14 backbone [38]* | | | | | | | |
| Candidate entities | 19.1 | 91.7 | 37.9 | 75.7 | 34.6 | 76.8 | – |
| + multimodal LLM correction | 24.8(+5.7) | 90.5(-1.2) | 52.3(+14.4) | 74.9(-0.8) | 38.3(+3.7) | 82.4(+5.6) | +13.9% |

LLM as a *verification and correction tool* rather than a *teacher* since directly using its predictions as targets result in poor performance.

**Zero-shot transfer to finegrained datasets.** In Tab. 2, we observe that the model trained on our proposed training dataset REW-47M transfers effectively to several finegrained datasets. The model trained on REW demonstrates superior transferability compared to the same model trained on captions or Entity-WebLI. This result shows the higher quality of the REW dataset.

**Using REW dataset as a memory base.** We explore the potential of our REW dataset when utilized as a memory base for visual matching in Tab. 3, and for retrieval-enhanced contrastive (RECO) training [18] in Appendix A.1.1. Each image in the memory is either associated with the candidate entity from text k-NN matching (as in Entity-WebLI [7]) or with the multimodal LLM corrected entity (our method for REW). In Tab. 3, we report the results of visual matching with two popular visual backbones [6, 41] and across six different finegrained visual entity recognition datasets for which we have the mapping from class label to Wikipedia entity. We see in Tab. 3 that our corrected entities lead to better visual matching performance across the board which suggest that they are better annotations, describing more accurately the content of the images. In fact, using our corrected entities boosts the performance in average by +10.5% relative improvement when considering CLIP [41] and by +13.9% relative improvement with DINOv2 [38].

### 4.3 Analysis and ablation study

Unless specified otherwise, models in this section are trained on the REW-5M dataset.

**Importance of entity verification and correction.** In Tab. 4, we train models with different source of annotations for the target entities. First, we observe that directly using the multimodal LLM raw output as target (which is akin to a distillation scenario) results in suboptimal performance (row 1). By inspecting qualitative results, our hypothesis is that this can be attributed to the LLM's tendency to produce hallucinations, overly generic answers, or outputs in an incorrect format (long descriptive captions instead of entities). Second, we validate in Tab. 4 the importance of the multimodal LLM correction step compared to using the candidate entities from text k-NN matching as in Entity-WebLI [7]: this strategy improves the performance of the resulting model by more than 6 points (row 3 versus row 2). Finally, providing additional contextual information to the multimodal LLM results in a substantial boost of 1.7 points (row 4 versus row 3), a benefit that we further illustrate through our qualitative analysis in Fig. 3. We note that the multimodal LLM refines the candidate entities 76% of the time, *i.e.* it validates the candidate entity 24% of the time (see Fig. 3c).

**Qualitative analysis on the importance of the multimodal LLM correction.** In Fig. 3, we identify two typical failure cases of the multimodal LLM correction step when it lacks access to Wikipedia and original caption metadata. The first case (see examples in Fig. 3a) involves the LLM making incorrect corrections by providing generic or hallucinated outputs. For example, it identifies an image of the *Bronte baths* as an *outdoor swimming pool* or it makes mistakes by recognizing incorrect plant or animal species. In contrast, the multimodal LLM with access to external metadata can get a hint about the correct entities by reading from the original caption. As a matter of fact, note that simply using the original captions as targets leads to suboptimal performance as shown by Caron et al. [7]. Intuitively, original captions are usually not in the form of an entity and can be noisy and not reflective of the visual entity of the image, as illustrated in the examples in Fig. 3b.

Table 4: **Importance of entity verification and correction.** We report HM top-1 accuracy on OVEN validation entity split. To isolate the effect of the entity target source, we train only with entity targets (*i.e.* only with loss $\mathcal{L}_{\text{Entity}}$). We do not perform seen finetuning and evaluate the models directly after pretraining. All models are trained with the *same pretraining images* and use the same architecture.

| Entity target source | Entity split (HM) |
|---|---|
| 1 Multimodal LLM raw output | 3.2 |
| 2 Candidate entity from text k-NN matching (as in Entity-WebLI [7]) | 6.7 |
| 3    + multimodal LLM correction *without* access to Wikipedia & original caption | 12.7 |
| 4    + multimodal LLM correction *with* access to Wikipedia & original caption | **14.1** |

(a) Incorrect candidate entity due to incorrect matching between original caption and Wikipedia entity names

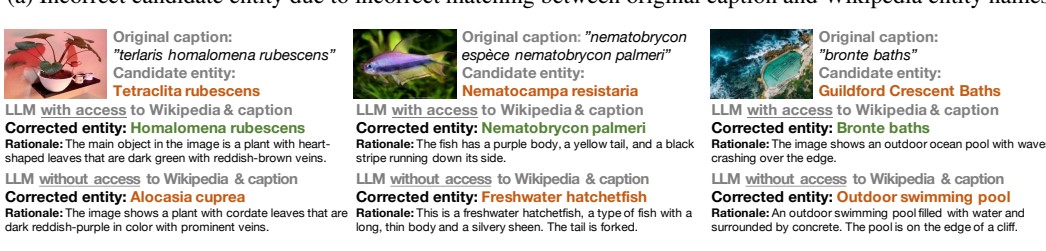

(b) Incorrect candidate entity due to irrelevant original caption

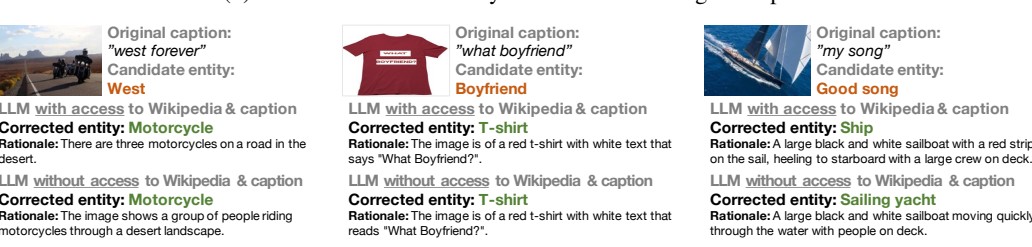

(c) Incorrect correction by multimodal LLM without access to Wikipedia external knowledge

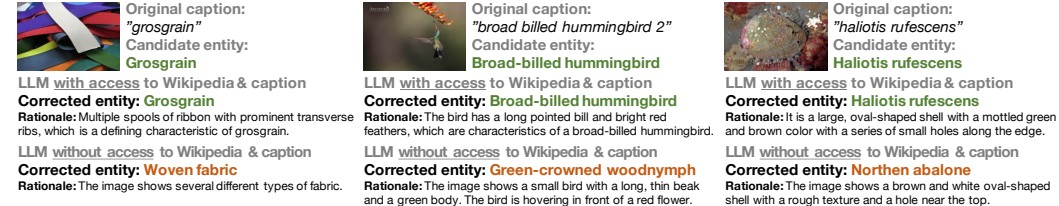

Figure 3: **Qualitative analysis of the importance of the entity verification and correction step.**

The second failure case of the multimodal LLM lacking access to metadata occurs when the model corrects information that it should not, likely due to a lack of knowledge about the entities involved (see examples in Fig. 3c). This issue is reflected in the LLM rationale, indicating a need for further external information about the candidate entities. By contrast, the model with access to Wikipedia can read about the visual attributes which are characteristic of the candidate entity and look for these in the image. This process is reflected in the LLM rationale: for example, the model with Wikipedia access to the *grosgrain* entity page validates this entity with the rationale: "*Multiple spools of ribbon with prominent transverse ribs, which is a defining characteristic of grosgrain*" while the model without Wikipedia seems to lack specific knowledge about *grosgrain*.

**Generating rationales and QA pairs with multimodal LLM.** In Tab. 5 (left), we probe the importance of providing external metadata to the LLM when generating rationales and question answer pairs. First, we observe in Tab. 5 (left) that the model without access to any metadata (row 1) has suboptimal performance. Incorporating external input, such as Wikipedia content and original captions, during the rationale generation process enhances the quality and pertinence of the rationale. Consequently, this leads to an improvement in performance (row 2). Importantly, this performance boost is even higher if the multimodal LLM can leverage the improved rationale when generating the QAs (row 3).

Table 5: **Ablation study: (left): Generating rationales and QA pairs with multimodal LLM. (right): Multi-task training with generated rationales and QAs.** We validate the robustness to the base image-caption dataset used by performing the latter ablation with both WebLI [9] and LAION [46]. We report HM top-1 accuracy on OVEN validation splits directly after REW training.

| | Metadata available to multimodal LLM | | | |
|---|---|---|---|---|
| | at rationale gen. | at QAs gen. | Entity | Query |
| 1 | ∅ | ∅ | 14.2 | 26.7 |
| 2 | Wiki & caption | ∅ | 14.7 | 27.6 |
| 3 | Wiki & caption | rationale | 15.5 | **29.0** |
| 4 | Wiki & caption | entity | **16.4** | 27.2 |
| 5 | Wiki & caption | entity + rationale | 16.0 | 28.2 |

| | | | | LLM-Refined Entity–. | | | |
|---|---|---|---|---|---|---|---|
| | | | | –WebLI | | –LAION | |
| | $\mathcal{L}_{\text{Entity}}$ | $\mathcal{L}_{\text{Rationale}}$ | $\mathcal{L}_{\text{QA}}$ | Entity | Query | Entity | Query |
| 1 | ✓ | | | 14.1 | 5.4 | 10.7 | 5.6 |
| 2 | ✓ | ✓ | | 14.6 | 6.7 | 11.4 | 6.9 |
| 3 | ✓ | | ✓ | 15.9 | 25.1 | 13.2 | 25.3 |
| 4 | ✓ | ✓ | ✓ | **16.0** | **28.2** | **13.4** | **28.2** |

In row 4, we notice that while it is important for the LLM to have access to the entity name during QA generation to improve entity split performance, it negatively impacts query split performance, as seen in the comparison between row 2 and row 4. This observation can be intuitively explained by the fact that when the LLM has access to the entity name, it is more likely to generate question/answer pairs focused on the main entity, rather than considering other attributes present in the image. However, these diverse attributes are often mentioned in the rationale, as demonstrated in the qualitative examples of Fig. 4 in Appendix. Therefore, having access to the rationale increases the variety of the generated QA pairs. Taking into account these observations, our default model (row 5) is configured to generate QAs by providing the model with access to both the entity and rationale. This approach aims to achieve a balance between strong performance in entity and query splits.

**Multi-task training with generated rationales and QAs.** In Tab. 5 (right), we confirm the importance of multi-task training with the various output types obtained from the multimodal LLM. We conduct this experiment using two different base image-caption datasets: WebLI [9] (our default) and the publicly available LAION dataset [46]. Our results demonstrate that both rationale and QA objectives contribute to the improved performance of our model. Intuitively, rationales help clarify the connection between entities and visual attributes, while QA training encourages the model to focus on multiple entities within the image. Moreover, both objectives enhance the model's language understanding, which we found to be important especially for the query split.

**Robustness to the base image-caption dataset.** Lastly, we observe that the results presented in Tab. 5 (right) are consistent across the different image-caption datasets used in our experiments. This suggests that our findings are robust and not specific to the WebLI dataset.

## 4.4 Results with open source models

Finally, we run an additional set of experiments with the open source PaliGemma [3] and Gemma 27B models [52]. We validate that the results are inline with the results when using the Gemini-Pro model [15]. Since Gemma lacks visual input processing, we replace direct image input with automatically generated captions. Specifically, we employ the open-source PaliGemma model to generate descriptive captions for each image using the prompt: *"Describe the visual attributes of this image."*. We then integrate these captions into the existing prompts outlined in Sec. A.3.1 by prepending the text: *"Here are the visual attributes of the image:"*.

We evaluate this approach on the 5M subset of WebLI and LAION. We compare the Gemma and Gemini-Pro variants of our method in Tab. 6 with the SOTA methods trained on the 5M subset of Entity-WebLI. We see in Tab. 6 that in all cases our method gives substantial improvements over the prior work GER-ALD [7] and GiT-Large trained on Entity-WebLI. In Tab. 8 of the Appendix, similarly to Tab. 5 (right) we evaluate the impact of the different losses when using the version of our method with open source models. We verify that conclusions are similar with private and open source models. As seen in Tab. 8, the main difference of performance between Gemini-Pro and Gemma variants comes from the QA loss. While Gemini-Pro has access to input images when generating QA pairs, Gemma generates QA pairs based on the PaliGemma caption (and rationale). This limits the variety of the generated QA pairs, resulting in a lower final accuracy.

Table 6: **Results with open source PaliGemma and Gemma models.** We report HM top-1 accuracy on OVEN validation entity split directly after REW training. We use a 5M subset for all the pretraining datasets.

|  | Pretraining dataset | Entity split (HM) | Query split (HM) |
|---|---|---|---|
| GiT-Large | Entity-WebLI [7] | 9.1 | 5.6 |
| GER-ALD [7] | Entity-WebLI [7] | 10.2 | – |
| GiT-Large | LLM-Refined Entity-LAION with Gemma | 11.6 | 23.4 |
| GiT-Large | LLM-Refined Entity-LAION with Gemini Pro | 13.4 | 28.2 |
| GiT-Large | LLM-Refined Entity-WebLI with Gemma | 14.2 | 24.3 |
| GiT-Large | LLM-Refined Entity-WebLI with Gemini Pro | 16.0 | 28.2 |

## 5 Discussion

**Limitations.** Our work relies on the use of multimodal LLMs with external knowledge bases, such as Wikipedia, for dataset creation and annotation. This dependence on using external data to prompt LLMs may limit the applicability of the approach in scenarios where such external knowledge is scarce or unavailable. Also, the proposed approach involves prompting the LLM to reason about candidate entity labels and generate additional metadata. This process is time-consuming and computationally expensive when considering multimodal LLMs with billions of parameters [15], which may limit the scalability of the approach to even larger datasets. A direction for future work could be to overcome some of the limitations of OVEN we find in our study (Appendix A.2) and create additional benchmarks for web-scale visual entity recognition.

**Conclusion.** We propose a novel methodology to curate a high-quality large-scale dataset, REW, for web-scale visual entity recognition using multimodal LLMs. Our approach significantly improves the quality of the dataset and bypasses the need for manual annotation. We also enrich the dataset with additional metadata, including question-answer pairs and rationales, generated by the LLM, which further improves the performance of the models. We achieve state-of-the-art performance on web-scale visual entity recognition tasks, highlighting the critical role of high-quality training data in this challenging domain. While our methodology for multimodal LLM-based data curation shows promise, we recognize significant opportunities for further enhancement. Integrating additional tools and external knowledge sources holds potential to improve its effectiveness. Furthermore, we anticipate that our approach can be broadly applicable to other visual tasks requiring extensive training data. Broader impact discussion is in Appendix A.4.

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

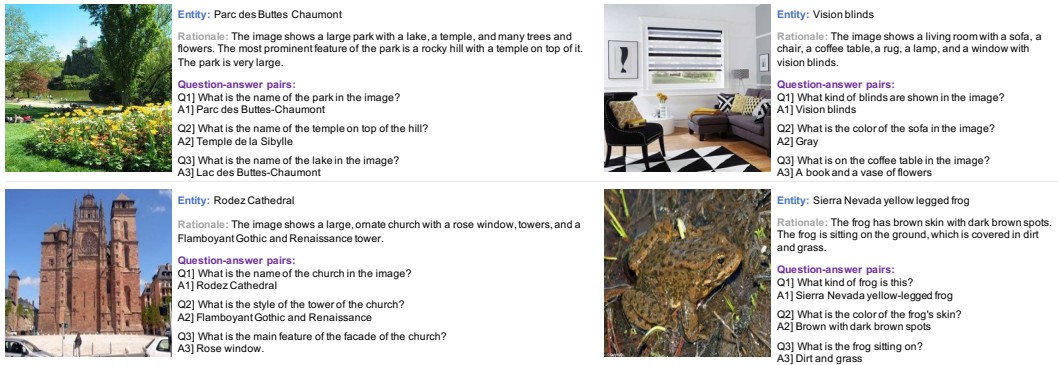

Figure 4: **Qualitative examples of entities, rationales and question-answer pairs** obtained with the multi-modal LLM. Our prompt encourage asking questions about diverse entities in the image.

Table 7: **Retrieval-enhanced contrastive training (RECO).** We compare three types of annotations for memory database: original caption, candidate entities or our multimodal LLM corrected entities. We report the *absolute* improvements over the CLIP baseline between parentheses as well as the *relative* improvement averaged across the datasets in the last column. For reference, we also include the results from Iscen et al. [18] but note that they use a memory base $20\times$ bigger (WebLI-1B).

| Memory dataset | Cars[20] | CUB[55] | ImNet[44] | Flowers[35] | Places[63] | Dogs[19] | Avg.relative $\Delta$ |
|---|---|---|---|---|---|---|---|
| CLIP-L/14 | 75.6 | 61.7 | 75.6 | 75.5 | 42.0 | 72.7 | – |
| *CLIP-L/14 + RECO* | | | | | | | |
| Captions - WebLI-1B [18] | 82.8 | 73.4 | 76.1 | 79.5 | 43.6 | 73.9 | +6.7% |
| Captions - WebLI-47M | 74.6(-1.0) | 73.8(+12.1) | 75.8(+0.2) | 77.8(+2.3) | 43.7(+1.7) | 73.4(+0.7) | +4.4% |
| Entity-WebLI | 76.0(+0.4) | 69.3(+7.6) | 75.8(+0.2) | 77.7(+2.2) | 43.5(+1.5) | 72.8(+0.1) | +3.3% |
| REW-47M (Ours) | 76.2(+0.6) | 72.6(+10.9) | 76.0(+0.4) | 81.6(+6.1) | 43.7(+1.7) | 73.8(+1.1) | **+5.4%** |

# A    Appendix and supplemental material

## A.1    Additional results

### A.1.1    A different application of our dataset: memory base for RECO.

We explore the potential of our dataset when utilized as a memory base for retrieval-enhanced contrastive training (RECO) [18] in Tab. 7. Each image in the memory is either associated with the original WebLI caption, with the candidate entity from text k-NN matching (as in Entity-WebLI [7]) or with the multimodal LLM corrected entity (our method REW-47M). In Tab. 7, we show that using the multimodal LLM to correct entity entities results in a relative improvement compared to CLIP of +5.4% on average across the six datasets considered. Using the original dataset captions instead results in a smaller relative improvement of +4.4%. Lastly, we see in Tab. 7 that we are able to achieve performance comparable to or even better than RECO's results on some datasets, despite using a memory base that is 20 times smaller. This finding suggests that a smaller amount of high-quality annotated data can be just as effective, if not more so, than a larger amount of data with lower annotation quality.

### A.1.2    Opensource models PaliGemma and Gemma

We see in Tab. 8 that the main difference of performance between Gemini-Pro and Gemma variants comes from the QA loss. While Gemini-Pro has access to input images when generating QA pairs, Gemma generates QA pairs based on the PaliGemma caption (and rationale). This limits the variety of the generated QA pairs, resulting in a lower final accuracy.

Table 8: **Analysis of the impact of multi-task training with generated rationales and QAs with private versus opensource models.** We perform this experiment with both WebLI [9] and LAION [46]. We report HM top-1 accuracy on OVEN validation splits directly after REW training.

| | | | LLM-Refined Entity–. | | | |
|---|---|---|---|---|---|---|
| | | | –WebLI | | –LAION | |
| $\mathcal{L}_{\text{Entity}}$ | $\mathcal{L}_{\text{Rationale}}$ | $\mathcal{L}_{\text{QA}}$ | Entity | Query | Entity | Query |
| *with private Gemini Pro model [15]* | | | | | | |
| ✓ | | | 14.1 | 5.4 | 10.7 | 5.6 |
| ✓ | ✓ | | 14.6 | 6.7 | 11.4 | 6.9 |
| ✓ | ✓ | ✓ | 16.0 | 28.2 | 13.4 | 28.2 |
| *with opensource PaliGemma [3] and Gemma [52] models* | | | | | | |
| ✓ | | | 11.9 | 5.8 | 9.5 | 9.4 |
| ✓ | ✓ | | 13.3 | 6.3 | 10.6 | 9.7 |
| ✓ | ✓ | ✓ | 14.2 | 24.3 | 11.6 | 23.4 |

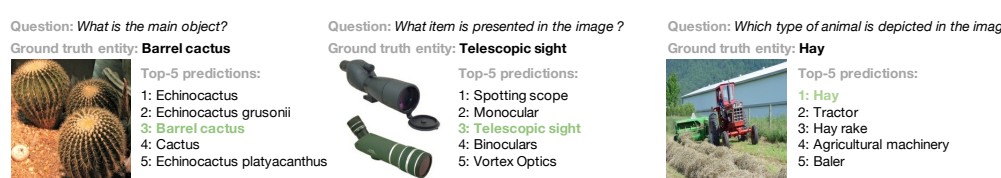

Figure 5: **Qualitative examples of suboptimal annotations in OVEN benchmark.** We show the input question, input image, OVEN ground truth entity as well as the top-5 predictions of our model.

## A.2 Limitations of the OVEN benchmark

The goal of this paper is to develop models capable of matching any given image-text query to an entity with a high precision. While the results in this work show that we improve upon the state of the art on the challenging web-scale visual entity recognition OVEN benchmark, we note that our best model still makes mistakes more than 70% of the time (see Tab. 1), which is far from optimal. While this shows that there is still a lot of room for improvement for further research in that direction, we provide a short analysis in this section of some limitations of the OVEN benchmark itself.

In Fig. 5, we show the top-5 predictions of our model for various examples from the OVEN validation set. We observe that, in some cases, our model's predictions are more accurate than the expected ground truth entities provided by the OVEN benchmark. This is because some of the positive entities in the OVEN validation or test set may have a better match within the negative entities. For instance, our model correctly predicts *Echinocactus grusonii* before the expected *Barrel cactus* entity, as the former is a more specific and accurate description of the cactus in the image. Similarly, our model predicts *spotting scope* instead of the expected *telescopic sight*, as the former is a more precise description of the visual entity in question.

To address these limitations, we suggest considering the top-k predictions when comparing the performance of different models. For example, our best-performing model's HM score increases dramatically from 29.6 to 57.0 when we considered the top-10 predictions instead of just the top-1 prediction.

Finally, the right side of Fig. 5 shows an example where the input question, *Which type of animal is depicted in the image?*, is not relevant to the input image, as there is no animal present. In this case, our model disregards the input question and outputs entities that are present in the image but are not animals such as *hay* or *tractor* for example. This raises the question of whether the model should prioritize the input question and provide a more relevant answer, even if it means potentially sacrificing accuracy. For instance, the model could output a response such as "There is no animal in the image," which would be more relevant to the user's query, but would not be strictly accurate in terms of entity recognition.

Table 9: **Statistical significance.** We report the harmonic mean (HM) of the seen and unseen splits (top-1 accuracy) on OVEN training seen categories for 5 different seeds. We use the small version of REW for this experiment. We report the mean and the standard deviation in the last row.

| Model | Pretraining data | Entity split | | | Query split | | |
|---|---|---|---|---|---|---|---|
| | | HM | seen | unseen | HM | seen | unseen |
| Seed 0 | REW (4.5M) | 16.2 | 19.6 | 13.7 | 29.1 | 32.6 | 26.3 |
| Seed 1 | REW (4.5M) | 15.6 | 19.0 | 13.3 | 27.9 | 32.3 | 24.5 |
| Seed 2 | REW (4.5M) | 16.3 | 19.6 | 13.9 | 28.0 | 32.7 | 24.5 |
| Seed 3 | REW (4.5M) | 16.1 | 19.7 | 13.6 | 27.5 | 32.4 | 23.8 |
| Seed 4 | REW (4.5M) | 15.9 | 19.3 | 13.5 | 28.7 | 32.6 | 25.5 |
| *Mean and standard deviation* | | $16.0_{\pm 0.2}$ | $19.4_{\pm 0.3}$ | $13.6_{\pm 0.2}$ | $28.2_{\pm 0.6}$ | $32.6_{\pm 0.1}$ | $24.9_{\pm 0.9}$ |

## A.3 Experimental Setting and Details

### A.3.1 Prompts

We input the image to the multimodal LLM as well as the following prompts.

**Entity correction and rationale generation prompt.** We include below our full prompt template for the entity correction and rationale generation process. We insert the original caption as **{original caption}**, the candidate entity as **{candidate entity}** and the Wikipedia page content summary of the candidate entity as **{Wikipedia summary}**.

---

*You are working on an entity recognition task.*
*Is this an image of **candidate entity**?*
*Your answer must be either 'YES' or 'NO'.*
*Here is the definition of **candidate entity**: **Wikipedia summary**.*
*If your answer is 'YES', you must use the definition of **candidate entity** to answer whether this is an image of a **candidate entity**.*
*If your answer is 'NO', you must use the caption of the image **original caption** to describe the main object in the image with the most specific English Wikipedia article title, where the response follows the format '@response@'. " You must then explain your answer by describing the visual attributes of the image.*
*If you answer is 'YES', your explanation MUST be based on the definition of **candidate entity**. If you answer is 'NO', your explanation MUST ONLY be based on the visual cues of the image, and it should NOT contain **candidate entity**.*
*Your explanation must be concise.*
*Your explanation MUST NOT exceed two sentences.*

---

**Question/answer pair generation prompt.** We include below our full prompt template for the question/answer pairs generation process. We insert the previously verified/corrected entity as **{entity}** and the previously generated rationale as **{rationale}**.

---

*You are working on a visual question answering task.*
*This is an image of **entity**.*
*Your rationale is the following: **rationale**.*
*Your task is to generate 3 question/answer pairs describing the visual attributes of this image.*
*The questions MUST be diverse and cover several entities of the image, including the main object in the image or image itself. The answers MUST be specific English Wikipedia article titles. The answers MUST be based on the visual content of the image and the provided rationale.*
*The format for the question/answer pairs is Q:<question> A:<answer>. The questions MUST NOT contain What is the main object in the image?.*

---

### A.3.2 Statistical significance of the experiments

We report the performance variance on our model trained on the small version of our dataset in order to assess the statistical significance of our results. We run this experiment five times with different random seeds. We perform the evaluation for each model separately and report in Tab. 9 the accuracy for each run in the different OVEN splits. We also report the mean over these five runs and the standard deviation.

### A.3.3 Compute resources for the experiments

Our models are trained on 256 TPUv3. The short schedule 200k steps training lasts for 15 hours while the longer 600k steps training lasts for 44 hours. The full scope of our research project involved more runs than what is reported in the paper. This is because we conducted preliminary explorations and experimented with various design choices that ultimately did not yield successful results. While these failed experiments are not presented in the tables of the paper, we believe that they are an important part of any research project and contributed to the development of our final approach.

### A.3.4 Implementation details

**Training and inference on REW dataset.** We use GiT-Large [56]: it consists of a visual encoder (CLIP-L/14 [41]) and a 6-layer text decoder with internal dimension $d = 768$. Following [7], the visual encoder is first pre-trained jointly on WebLI-100M [9] and Conceptual Captions-12M [47] while the decoder is randomly initialized. We use batch size of 4096, learning rate of $1e^{-5}$ for the visual encoder and $1e^{-4}$ for the decoder, label smoothing of $0.2$ and no weight decay. We use AdamW optimizer [27] and a cosine learning rate schedule with final learning rate of $0$. We use standard inception crop data augmentation for the images. We set the maximum decoding length to 32 tokens and the maximum number of context tokens to 32 tokens as well. We find that 32 tokens is enough to comprehensively tokenize all the OVEN input questions as well as all the Wikipedia entity names. The decoding beam size is set to 30.

**Finetuning on OVEN train set.** We finetune models on OVEN training set for 10,000 steps with a batch size of 256. Note that we equally balance the query and entity split contribution during finetuning. We choose the learning rate out of three values ($1e^{-7}$, $3e^{-7}$, $1e^{-6}$) based on the HM performance on the OVEN validation set. Label smoothing is set at $0.1$. Note that the finetuning schedule is relatively short (10,000 steps) because we observe that long finetuning (or equivalently, using a large learning rate) causes the model to forget about the unseen categories.

### A.4 Broader impacts discussion

**Potential positive societal impacts.** Our proposed methodology for curating a high-quality, large-scale dataset for web-scale visual entity recognition using multimodal LLMs has the potential to significantly impact the field of computer vision. By improving the quality of training data, we can enhance the performance of models for various entity-aware visual understanding tasks, including info-seeking VQA or News content understanding, leading to more accurate and reliable results. Furthermore, our approach has the potential to democratize the process of dataset creation, reducing the time and resources required for manual annotation. This can enable researchers and practitioners with limited resources to create high-quality datasets.

**Potential negative societal impacts.** However, it is essential to consider the potential risks and ethical implications of our work. The use of LLMs for dataset creation and annotation raises concerns about bias and fairness, as the models may reflect and perpetuate the biases present in the data they were trained on. Therefore, it is important to carefully evaluate and mitigate any potential biases in the datasets created using our approach. Additionally, the use of large-scale image-captions such as WebLI [9] or LAION [46] databases for entity recognition raises concerns about privacy and data protection, as the images and captions used for training may contain sensitive information. Therefore, it is essential to ensure that the datasets created using our approach are compliant with relevant privacy and data protection regulations and that appropriate measures are taken to protect the privacy and security of the data.

In summary, our work has the potential to significantly impact the field of computer vision and enable the development of more accurate and reliable models for various entity-aware visual understanding

tasks. However, it is essential to carefully consider and address the potential risks and ethical implications of our work to ensure that it is used for the benefit of all.

## A.5 Class label to Wikipedia entity mappings

We selected five finegrained datasets for evaluation because their class categories align with a subset of the Wikipedia entities that our model is trained to recognize. We obtained a preliminary mapping of class labels to Wikipedia entities from the authors of OVEN and then improved it through a careful manual review.

**Oxford Flowers [35].** We use the following **class name** to Wikipedia entity name mapping:
[**pink primrose**: oenothera speciosa; **hard-leaved pocket orchid**: paphiopedilum micranthum; **canterbury bells**: campanula medium; **sweet pea**: sweet pea; **english marigold**: calendula officinalis; **tiger lily**: lilium lancifolium; **moon orchid**: phalaenopsis amabilis; **bird of paradise**: strelitzia; **monkshood**: aconitum; **globe thistle**: echinops; **snapdragon**: antirrhinum; **colt's foot**: tussilago; **king protea**: protea cynaroides; **spear thistle**: cirsium vulgare; **yellow iris**: iris pseudacorus; **globeflower**: trollius europaeus; **purple coneflower**: echinacea purpurea; **peruvian lily**: alstroemeria; **balloon flower**: platycodon; **giant white arum lily**: zantedeschia; **fire lily**: lilium bulbiferum; **pincushion flower**: scabiosa; **fritillary**: fritillaria; **red ginger**: alpinia purpurata; **grape hyacinth**: muscari; **corn poppy**: papaver rhoeas; **prince of wales feathers**: amaranthus hypochondriacus; **stemless gentian**: stemless gentian; **artichoke**: artichoke; **sweet william**: dianthus barbatus; **carnation**: dianthus caryophyllus; **garden phlox**: phlox paniculata; **love in the mist**: nigella damascena; **mexican aster**: cosmos bipinnatus; **alpine sea holly**: eryngium alpinum; **ruby-lipped cattleya**: cattleya labiata; **cape flower**: nerine bowdenii; **great masterwort**: astrantia major; **siam tulip**: curcuma alismatifolia; **lenten rose**: hellebore; **barbeton daisy**: gerbera jamesonii; **daffodil**: narcissus (plant); **sword lily**: gladiolus; **poinsettia**: poinsettia; **bolero deep blue**: eustoma russellianum; **wallflower**: erysimum; **marigold**: tagetes; **buttercup**: ranunculus; **oxeye daisy**: leucanthemum vulgare; **common dandelion**: taraxacum officinale; **petunia**: petunia; **wild pansy**: viola tricolor; **primula**: primula; **sunflower**: common sunflower; **pelargonium**: pelargonium; **bishop of llandaff**: dahlia 'bishop of llandaff'; **gaura**: gaura; **geranium**: geranium; **orange dahlia**: tithonia rotundifolia; **pink-yellow dahlia?**: dahlia; **cautleya spicata**: cautleya spicata; **japanese anemone**: eriocapitella japonica; **black-eyed susan**: rudbeckia hirta; **silverbush**: convolvulus cneorum; **californian poppy**: eschscholzia californica; **osteospermum**: osteospermum; **spring crocus**: crocus vernus; **bearded iris**: iris (plant); **windflower**: anemonoides blanda; **tree poppy**: romneya; **gazania**: gazania; **azalea**: azalea; **water lily**: nymphaeaceae; **rose**: rose; **thorn apple**: datura; **morning glory**: morning glory; **passion flower**: passiflora; **lotus**: nelumbo nucifera; **toad lily**: tricyrtis hirta; **anthurium**: anthurium; **frangipani**: plumeria; **clematis**: clematis; **hibiscus**: hibiscus; **columbine**: aquilegia; **desert-rose**: adenium obesum; **tree mallow**: malva arborea; **magnolia**: magnolia; **cyclamen**: cyclamen; **watercress**: watercress; **canna lily**: canna (plant); **hippeastrum**: hippeastrum; **bee balm**: monarda; **ball moss**: wallisia; **foxglove**: digitalis; **bougainvillea**: bougainvillea; **camellia**: camellia; **mallow**: althaea officinalis; **mexican petunia**: ruellia simplex; **bromelia**: bromelia; **blanket flower**: gaillardia; **trumpet creeper**: campsis radicans; **blackberry lily**: iris domestica].

**Sun397 [57].** We use the following **class name** to Wikipedia entity name mapping:
[**abbey**: abbey; **airplane cabin**: aircraft cabin; **airport terminal**: airport terminal; **alley**: alley; **amphitheater**: amphitheatre; **amusement arcade**: amusement arcade; **amusement park**: amusement park; **anechoic chamber**: anechoic chamber; **apartment building/outdoor**: apartment; **apse/indoor**: apse; **aquarium**: aquarium; **aqueduct**: aqueduct (bridge); **arch**: arch; **archive**: archive; **arrival gate/outdoor**: airport apron; **art gallery**: art gallery; **art school**: art school; **art studio**: studio; **assembly line**: assembly line; **athletic field/outdoor**: pitch (sports field); **atrium/public**: atrium (architecture); **attic**: attic; **auditorium**: auditorium; **auto factory**: automotive industry; **badlands**: badlands; **badminton court/indoor**: badminton; **baggage claim**: baggage reclaim; **bakery/shop**: bakery; **balcony/exterior**: balcony; **balcony/interior**: mezzanine; **ball pit**: ball pit; **ballroom**: ballroom; **bamboo forest**: bamboo; **banquet hall**: banquet hall; **bar**: bar (establishment); **barn**: barn; **barndoor**: barnyard; **baseball field**: baseball field; **basement**: basement; **basilica**: basilica; **basketball court/outdoor**: basketball court; **bathroom**: bathroom; **batters box**: batting (baseball); **bayou**: bayou; **bazaar/indoor**: bazaar; **bazaar/outdoor**: marketplace; **beach**: beach; **beauty salon**: beauty salon; **bedroom**: bedroom; **berth**: bunk bed; **biology laboratory**: laboratory; **bistro/indoor**: bistro; **boardwalk**: boardwalk; **boat deck**: deck (ship); **boathouse**: boathouse; **bookstore**: bookselling; **booth/indoor**: convention (meeting); **botanical garden**: botanical garden;

**bow window/indoor**: window; **bow window/outdoor**: bow window; **bowling alley**: bowling alley; **boxing ring**: boxing ring; **brewery/indoor**: brewery; **bridge**: bridge; **building facade**: façade; **bullring**: bullring; **burial chamber**: chamber tomb; **bus interior**: bus; **butchers shop**: butcher; **butte**: butte; **cabin/outdoor**: log cabin; **cafeteria**: cafeteria; **campsite**: campsite; **campus**: campus; **canal/natural**: canal (garden history); **canal/urban**: canal; **candy store**: confectionery store; **canyon**: canyon; **car interior/backseat**: car seat; **car interior/frontseat**: bucket seat; **carrousel**: carousel; **casino/indoor**: casino; **castle**: castle; **catacomb**: catacombs; **cathedral/indoor**: sanctuary; **cathedral/outdoor**: cathedral; **cavern/indoor**: cave; **cemetery**: cemetery; **chalet**: chalet; **cheese factory**: creamery; **chemistry lab**: chemistry; **chicken coop/indoor**: chicken; **chicken coop/outdoor**: poultry farming; **childs room**: child care; **church/indoor**: nave; **church/outdoor**: church (building); **classroom**: classroom; **clean room**: cleanroom; **cliff**: cliff; **cloister/indoor**: cloister; **closet**: closet; **clothing store**: clothes shop; **coast**: coast; **cockpit**: cockpit; **coffee shop**: coffeehouse; **computer room**: computer lab; **conference center**: convention center; **conference room**: conference hall; **construction site**: construction; **control room**: control room; **control tower/outdoor**: air traffic control; **corn field**: harvest; **corral**: pen (enclosure); **corridor**: hallway; **cottage garden**: cottage garden; **courthouse**: courthouse; **courtroom**: courtroom; **courtyard**: courtyard; **covered bridge/exterior**: covered bridge; **creek**: stream; **crevasse**: crevasse; **crosswalk**: pedestrian crossing; **cubicle/office**: cubicle; **dam**: dam; **delicatessen**: delicatessen; **dentists office**: dentistry; **desert/sand**: desert; **desert/vegetation**: deserts and xeric shrublands; **diner/indoor**: diner; **diner/outdoor**: diner; **dinette/home**: table (furniture); **dinette/vehicle**: recreational vehicle; **dining car**: dining car; **dining room**: dining room; **discotheque**: nightclub; **dock**: dock; **doorway/outdoor**: door; **dorm room**: dormitory; **driveway**: driveway; **driving range/outdoor**: driving range; **drugstore**: pharmacy (shop); **electrical substation**: electrical substation; **elevator/door**: automatic door; **elevator/interior**: elevator; **elevator shaft**: shaft (mechanical engineering); **engine room**: engine room; **escalator/indoor**: escalator; **excavation**: excavator; **factory/indoor**: factory; **fairway**: lawn; **fastfood restaurant**: fast-food restaurant; **field/cultivated**: field (agriculture); **field/wild**: wild field; **fire escape**: fire escape; **fire station**: fire station; **firing range/indoor**: shooting range; **fishpond**: fishpond; **florist shop/indoor**: floristry; **food court**: food court; **forest/broadleaf**: forest; **forest/needleleaf**: conifer; **forest path**: forest track; **forest road**: agricultural road; **formal garden**: formal garden; **fountain**: fountain; **galley**: galley (kitchen); **game room**: game room; **garage/indoor**: garage (residential); **garbage dump**: waste container; **gas station**: filling station; **gazebo/exterior**: gazebo; **general store/indoor**: retail; **general store/outdoor**: general store; **gift shop**: gift shop; **golf course**: golf course; **greenhouse/indoor**: garden; **greenhouse/outdoor**: greenhouse; **gymnasium/indoor**: gymnasium (school); **hangar/indoor**: airship hangar; **hangar/outdoor**: hangar; **harbor**: harbor; **hayfield**: hay; **heliport**: heliport; **herb garden**: kitchen garden; **highway**: highway; **hill**: hill; **home office**: small office/home office; **hospital**: hospital; **hospital room**: room service; **hot spring**: hot spring; **hot tub/outdoor**: hot tub; **hotel/outdoor**: hotel; **hotel room**: suite (hotel); **house**: house; **hunting lodge/outdoor**: hunting and shooting in the united kingdom; **ice cream parlor**: ice cream parlor; **ice floe**: ice floe; **ice shelf**: ice shelf; **ice skating rink/indoor**: ice skating; **ice skating rink/outdoor**: ice rink; **iceberg**: iceberg; **igloo**: igloo; **industrial area**: industrial district; **inn/outdoor**: inn; **islet**: islet; **jacuzzi/indoor**: jacuzzi; **jail/indoor**: prison; **jail cell**: prison cell; **jewelry shop**: jewellery store; **kasbah**: kasbah; **kennel/indoor**: kennel; **kennel/outdoor**: dog crate; **kindergarden classroom**: kindergarten; **kitchen**: kitchen; **kitchenette**: kitchenette; **labyrinth/outdoor**: labyrinth; **lake/natural**: lake; **landfill**: landfill; **landing deck**: flight deck; **laundromat**: self-service laundry; **lecture room**: lecture room; **library/indoor**: library; **library/outdoor**: public library; **lido deck/outdoor**: lido; **lift bridge**: vertical-lift bridge; **lighthouse**: lighthouse; **limousine interior**: limousine; **living room**: living room; **lobby**: lobby (room); **lock chamber**: lock (water navigation); **locker room**: changing room; **mansion**: mansion; **manufactured home**: manufactured housing; **market/indoor**: grocery store; **market/outdoor**: farmers' market; **marsh**: marsh; **martial arts gym**: martial arts; **mausoleum**: mausoleum; **medina**: medina quarter; **moat/water**: moat; **monastery/outdoor**: monastery; **mosque/indoor**: islam; **mosque/outdoor**: mosque; **motel**: motel; **mountain**: mountain; **mountain snowy**: glacier; **movie theater/indoor**: movie theater; **museum/indoor**: museum; **music store**: music store; **music studio**: recording studio; **nuclear power plant/outdoor**: nuclear power plant; **nursery**: nursery (room); **oast house**: oast house; **observatory/outdoor**: observatory; **ocean**: ocean; **office**: desk; **office building**: office; **oil refinery/outdoor**: oil refinery; **oilrig**: oil rig; **operating room**: operating theater; **orchard**: orchard; **outhouse/outdoor**: outhouse; **pagoda**: pagoda; **palace**: palace; **pantry**: pantry; **park**: park; **parking garage/indoor**: parking space; **parking garage/outdoor**: multistorey car park; **parking lot**: parking lot; **parlor**: parlour; **pasture**: pasture; **patio**: patio; **pavilion**: pavilion; **pharmacy**: pharmacy; **phone**

**booth**: telephone booth; **physics laboratory**: physics; **picnic area**: picnic; **pilothouse/indoor**: bridge (nautical); **planetarium/outdoor**: planetarium; **playground**: playground; **playroom**: family room; **plaza**: town square; **podium/indoor**: lectern; **podium/outdoor**: podium; **pond**: pond; **poolroom/establishment**: billiard room; **poolroom/home**: billiard table; **power plant/outdoor**: power station; **promenade deck**: promenade deck; **pub/indoor**: pub; **pulpit**: pulpit; **putting green**: greenskeeper; **racecourse**: horse racing; **raceway**: race track; **raft**: raft; **railroad track**: railway track; **rainforest**: rainforest; **reception**: front office; **recreation room**: recreation room; **residential neighborhood**: residential area; **restaurant**: restaurant; **restaurant kitchen**: chef; **restaurant patio**: outdoor dining; **rice paddy**: paddy field; **riding arena**: riding hall; **river**: river; **rock arch**: natural arch; **rope bridge**: simple suspension bridge; **ruin**: ruins; **runway**: runway; **sandbar**: shoal; **sandbox**: sandpit; **sauna**: sauna; **schoolhouse**: school; **sea cliff**: cliffed coast; **server room**: server room; **shed**: shed; **shoe shop**: shoemaking; **shopfront**: storefront; **shopping mall/indoor**: shopping mall; **shower**: shower; **skatepark**: skatepark; **ski lodge**: ski lodge; **ski resort**: ski resort; **ski slope**: alpine skiing; **sky**: sky; **skyscraper**: skyscraper; **slum**: slum; **snowfield**: snow field; **squash court**: squash (sport); **stable**: stable; **stadium/baseball**: ballpark; **stadium/football**: stadium; **stage/indoor**: stage (theatre); **staircase**: stairs; **street**: street; **subway interior**: public transport; **subway station/platform**: metro station; **supermarket**: supermarket; **sushi bar**: sushi; **swamp**: swamp; **swimming pool/indoor**: swimming pool; **swimming pool/outdoor**: lido; **synagogue/indoor**: bema; **synagogue/outdoor**: synagogue; **television studio**: television studio; **temple/east asia**: temple; **temple/south asia**: hindu temple; **tennis court/indoor**: carpet court; **tennis court/outdoor**: tennis court; **tent/outdoor**: tent; **theater/indoor procenium**: proscenium; **theater/indoor seats**: theatre; **thriftshop**: charity shop; **throne room**: throne room; **ticket booth**: box office; **toll plaza**: toll road; **topiary garden**: topiary; **tower**: tower; **toyshop**: toy store; **track/outdoor**: running track; **train railway**: rail transport; **train station/platform**: train station; **tree farm**: tree farm; **tree house**: tree house; **trench**: trench; **underwater/coral reef**: coral reef; **utility room**: utility room; **valley**: valley; **van interior**: van; **vegetable garden**: vegetable farming; **veranda**: veranda; **veterinarians office**: veterinarian; **viaduct**: viaduct; **videostore**: video rental shop; **village**: village; **vineyard**: vineyard; **volcano**: volcano; **volleyball court/indoor**: volleyball; **volleyball court/outdoor**: beach volleyball; **waiting room**: waiting room; **warehouse/indoor**: warehouse; **water tower**: water tower; **waterfall/block**: rapids; **waterfall/fan**: waterfall; **waterfall/plunge**: plunge pool; **watering hole**: depression (geology); **wave**: wave; **wet bar**: wet bar; **wheat field**: wheat fields; **wind farm**: wind farm; **windmill**: windmill; **wine cellar/barrel storage**: barrel; **wine cellar/bottle storage**: wine cellar; **wrestling ring/indoor**: wrestling ring; **yard**: yard (land); **youth hostel**: hostel].

**Food101 [5].** We use the following **class name** to Wikipedia entity name mapping:
[**apple pie**: apple pie; **baby back ribs**: pork ribs; **baklava**: baklava; **beef carpaccio**: carpaccio; **beef tartare**: steak tartare; **beet salad**: vinegret; **beignets**: beignet; **bibimbap**: bibimbap; **bread pudding**: bread pudding; **breakfast burrito**: breakfast burrito; **bruschetta**: bruschetta; **caesar salad**: caesar salad; **cannoli**: cannoli; **caprese salad**: caprese salad; **carrot cake**: carrot cake; **ceviche**: ceviche; **cheesecake**: cheesecake; **cheese plate**: cheese; **chicken curry**: chicken curry; **chicken quesadilla**: quesadilla; **chicken wings**: chicken as food; **chocolate cake**: chocolate cake; **chocolate mousse**: mousse; **churros**: churro; **clam chowder**: clam chowder; **club sandwich**: club sandwich; **crab cakes**: crab cake; **creme brulee**: crème brûlée; **croque madame**: croque monsieur; **cup cakes**: cupcake; **deviled eggs**: deviled egg; **donuts**: doughnut; **dumplings**: dumpling; **edamame**: edamame; **eggs benedict**: eggs benedict; **escargots**: snails as food; **falafel**: falafel; **filet mignon**: filet mignon; **fish and chips**: fish and chips; **foie gras**: foie gras; **french fries**: french fries; **french onion soup**: french onion soup; **french toast**: french toast; **fried calamari**: squid as food; **fried rice**: fried rice; **frozen yogurt**: frozen yogurt; **garlic bread**: garlic bread; **gnocchi**: gnocchi; **greek salad**: greek salad; **grilled cheese sandwich**: grilled cheese; **grilled salmon**: list of potato chip brands; **guacamole**: guacamole; **gyoza**: jiaozi; **hamburger**: hamburger; **hot and sour soup**: hot and sour soup; **hot dog**: hot dog; **huevos rancheros**: huevos rancheros; **hummus**: hummus; **ice cream**: ice cream; **lasagna**: lasagna; **lobster bisque**: bisque (food); **lobster roll sandwich**: lobster roll; **macaroni and cheese**: macaroni and cheese; **macarons**: macaron; **miso soup**: miso soup; **mussels**: mussel; **nachos**: nachos; **omelette**: omelette; **onion rings**: onion ring; **oysters**: oyster; **pad thai**: pad thai; **paella**: paella; **pancakes**: pancake; **panna cotta**: panna cotta; **peking duck**: peking duck; **pho**: pho; **pizza**: pizza; **pork chop**: pork chop; **poutine**: poutine; **prime rib**: standing rib roast; **pulled pork sandwich**: pulled pork; **ramen**: ramen; **ravioli**: ravioli; **red velvet cake**: red velvet cake; **risotto**: risotto; **samosa**: samosa; **sashimi**: sashimi; **scallops**: scallop; **seaweed salad**: wakame; **shrimp and grits**: shrimp and grits; **spaghetti bolognese**: bolognese sauce; **spaghetti carbonara**: carbonara;

spring rolls: spring roll; **steak**: steak; **strawberry shortcake**: shortcake; **sushi**: sushi; **tacos**: taco; **takoyaki**: takoyaki; **tiramisu**: tiramisu; **tuna tartare**: tuna; **waffles**: waffle].

**FGVC-Aircraft [29].** We use the following **class name** to Wikipedia entity name mapping: [**Boeing 707**: boeing 707; **Boeing 727**: boeing 727; **Boeing 737**: boeing 737; **Boeing 747**: boeing 747; **Boeing 757**: boeing 757; **Boeing 767**: boeing 767; **Boeing 777**: boeing 777; **A300**: airbus a300; **A310**: airbus a310; **A320**: airbus a320 family; **A330**: airbus a330; **A340**: airbus a340; **A380**: airbus a380; **ATR-42**: atr 42; **ATR-72**: atr 72; **An-12**: antonov an-12; **BAE 146**: british aerospace 146; **BAE-125**: british aerospace 125; **Beechcraft 1900**: beechcraft 1900; **Boeing 717**: boeing 717; **C-130**: lockheed c-130 hercules; **C-47**: douglas c-47 skytrain; **CRJ-200**: bombardier crj100/200; **CRJ-700**: bombardier crj700 series; **Cessna 172**: cessna 172; **Cessna 208**: cessna 208 caravan; **Cessna Citation**: cessna citation family; **Challenger 600**: bombardier challenger 600 series; **DC-10**: mcdonnell douglas dc-10; **DC-3**: douglas dc-3; **DC-6**: douglas dc-6; **DC-8**: douglas dc-8; **DC-9**: mcdonnell douglas dc-9; **DH-82**: de havilland tiger moth; **DHC-1**: de havilland canada dhc-1 chipmunk; **DHC-6**: de havilland canada dhc-6 twin otter; **Dash 8**: de havilland canada dash 8; **DR-400**: robin dr400; **Dornier 328**: dornier 328; **Embraer E-Jet**: embraer e-jet family; **EMB-120**: embraer emb 120 brasilia; **Embraer ERJ 145**: embraer erj family; **Embraer Legacy 600**: embraer legacy 600; **Eurofighter Typhoon**: eurofighter typhoon; **F-16**: general dynamics f-16 fighting falcon; **F/A-18**: mcdonnell douglas f/a-18 hornet; **Falcon 2000**: dassault falcon 2000; **Falcon 900**: dassault falcon 900; **Fokker 100**: fokker 100; **Fokker 50**: fokker 50; **Fokker 70**: fokker 70; **Global Express**: bombardier global express; **Gulfstream**: gulfstream aerospace; **Hawk T1**: bae systems hawk; **Il-76**: ilyushin il-76; **L-1011**: lockheed l-1011 tristar; **MD-11**: mcdonnell douglas md-11; **MD-80**: mcdonnell douglas md-80; **MD-90**: mcdonnell douglas md-90; **Metroliner**: fairchild swearingen metroliner; **King Air**: beechcraft king air; **PA-28**: piper pa-28 cherokee; **SR-20**: cirrus sr20; **Saab 2000**: saab 2000; **Saab 340**: saab 340; **Spitfire**: supermarine spitfire; **Tornado**: panavia tornado; **Tu-134**: tupolev tu-134; **Tu-154**: tupolev tu-154; **Yak-42**: yakovlev yak-42].

**Sports100 [17].** We use the following **class name** to Wikipedia entity name mapping: [**air hockey**: air hockey; **ampute football**: amputee football; **archery**: archery; **arm wrestling**: arm wrestling; **axe throwing**: axe throwing; **balance beam**: balance beam; **barell racing**: barrel racing; **baseball**: baseball; **basketball**: basketball; **baton twirling**: baton twirling; **bike polo**: cycle polo; **billiards**: pool (cue sports); **bmx**: bmx; **bobsled**: bobsleigh; **bowling**: bowling; **boxing**: boxing; **bull riding**: bull riding; **bungee jumping**: bungee jumping; **canoe slamon**: canoe slalom; **cheerleading**: cheerleading; **chuckwagon racing**: chuckwagon racing; **cricket**: cricket; **croquet**: croquet; **curling**: curling; **disc golf**: disc golf; **fencing**: fencing; **field hockey**: field hockey; **figure skating men**: figure skating; **figure skating pairs**: pair skating; **figure skating women**: single skating; **fly fishing**: fly fishing; **football**: football; **formula 1 racing**: formula one racing; **frisbee**: frisbee; **gaga**: gaga; **giant slalom**: giant slalom; **golf**: golf; **hammer throw**: hammer throw; **hang gliding**: hang gliding; **harness racing**: harness racing; **high jump**: high jump; **hockey**: underwater ice hockey; **horse jumping**: show jumping; **horse racing**: horse racing; **horseshoe pitching**: horseshoes; **hurdles**: hurdling; **hydroplane racing**: hydroplane racing; **ice climbing**: ice climbing; **ice yachting**: iceboat; **jai alai**: jai alai; **javelin**: javelin; **jousting**: jousting; **judo**: judo; **lacrosse**: lacrosse; **log rolling**: logrolling (sport); **luge**: luge; **motorcycle racing**: motorcycle racing; **mushing**: mushing; **nascar racing**: nascar racing; **olympic wrestling**: wrestling; **parallel bar**: parallel bars; **pole climbing**: pole climbing; **pole dancing**: pole dance; **pole vault**: pole vault; **polo**: polo; **pommel horse**: pommel horse; **rings**: rings (gymnastics); **rock climbing**: rock climbing; **rollerblade racing**: inline skating; **roller derby**: roller derby; **rowing**: rowing; **rugby**: rugby union; **sailboat racing**: sailing (sport); **shot put**: shot put; **shuffleboard**: shuffleboard; **sidecar racing**: sidecar; **ski jumping**: ski jumping; **skydiving**: parachuting; **sky surfing**: skysurfing; **snow boarding**: snowboarding; **snowmobile racing**: snowmobile; **speed skating**: speed skating; **steer wrestling**: steer wrestling; **sumo wrestling**: sumo; **surfing**: surfing; **swimming**: swimming; **table tennis**: table tennis; **tennis**: tennis; **track bicycle**: track cycling; **trapeze**: trapeze; **tug of war**: tug of war; **ultimate**: ultimate (sport); **uneven bars**: uneven bars; **volleyball**: volleyball; **water cycling**: aqua cycling; **water polo**: water polo; **weightlifting**: weightlifting; **wheelchair basketball**: wheelchair basketball; **wheelchair racing**: wheelchair racing; **wingsuit flying**: wingsuit flying].

