# OpenReview forum: "Web-Scale Visual Entity Recognition: An LLM-Driven Data Approach"
_NeurIPS.cc/2024/Conference — NeurIPS 2024 poster_

### Official Review · Reviewer_9wiz · 2024-06-14

**Soundness:** 4
**Presentation:** 4
**Contribution:** 4
**Rating:** 8
**Confidence:** 5

**Summary:**

This work focuses on an important and realistic visual task called visual entity recognition (image, query --> entity name).
This work proposes a LLM driven technique to generate and refine large scale training data by modifying incorrectly labeled entity names from Entity-WebLI. Several errors are identified in Entity-WebLI and this work proposes several techniques to address them.
The experiment on OVEN and OOD datasets confirmed the usefulness of the new dataset REW and ablation studies show important design decisions contributing to the performance.

**Strengths:**

- Proposing LLM driven approach to refine labels in a visual entity recognition training dataset. Identifying several issues in the existing dataset creation pipeline and show LLM can help to correct them using additional context from meta data or wikipedia caption with ablation studies.
- Thorough experiments and ablation studies confirm the design choices such as QA, rationale generation for performance improvement.

**Weaknesses:**

- I don't see major weakness of this work except using LLM for data generation might be technically limited. However, I believe the techniques used to correct entity labels and findings are useful and novel for large scale entity centric caption creation.
- releasing data and code: Although the data will not be released, I think the reproducibility of experiments are supported by the experiments with 5 random seeds and also using open-sourced data such as LAION. The community will benefit from reproducing such large-scale entity recognition image data by using LAION and multimodal LLM such as LLaVA.

**Questions:**

- I wonder how the REW benchmark generalize to OOD entities of OVEN (landmarks, animals, plants, car, aircraft, etc).
Can you show some examples of the model predictions on visual entity categories not covered by the OVEN benchmark such as artwork, shopping items, etc.

- It seems using rational data improves query split in OVEN. I wonder if the authors have any intuition about this.

---

> ### Author Rebuttal · Authors · 2024-08-06
>
> **releasing data and code: Although the data will not be released, I think the reproducibility of experiments are supported by the experiments with 5 random seeds and also using open-sourced data such as LAION. The community will benefit from reproducing such large-scale entity recognition image data by using LAION and multimodal LLM such as LLaVA.**
>
> We intend to release LAION annotations. We also intend to release the implementation of the open source Gemma variant presented in the “General response” of this rebuttal.
>
>
> **I wonder how the REW benchmark generalize to OOD entities of OVEN (landmarks, animals, plants, car, aircraft, etc). Can you show some examples of the model predictions on visual entity categories not covered by the OVEN benchmark such as artwork, shopping items, etc.**
>
> We show results for shopping items, such as chairs, lamps and bicycles from the Stanford Online Products dataset with our model in the rebuttal PDF. We can observe that our model is able to predict the correct fine-grained classes, as the corresponding entities are contained in the 6M entity list of Wikipedia, which is much more extensive than the OVEN entities.
>
> **It seems using rational data improves query split in OVEN. I wonder if the authors have any intuition about this.**
>
> We believe that the rationales help clarify the connections between entities and visual attributes. Furthermore, providing rationales as additional supervision improves the model's language processing capabilities, which is important for deciphering complex text queries. These improvements are particularly valuable for VQA tasks, where an understanding of both visual and linguistic information is essential for accurate question interpretation and answer generation.

---

> > ### Comment · Reviewer_9wiz · 2024-08-13
> >
> > Thanks for the REW benchmark experiments and sharing LAION annotations with Gemma variant implementation. I think 8 is a fair evaluation of this work.

---

### Official Review · Reviewer_LX9i · 2024-06-30

**Soundness:** 3
**Presentation:** 3
**Contribution:** 3
**Rating:** 7
**Confidence:** 4

**Summary:**

The work proposes a data-centric approach, that leverages MLLMs to first verify the correspondence of the existing pre-training dataset (Entity-WebLI) and correct errors, ask them to produce rationales, and generate novel, query split-oriented QA pairs to train a language model. The refined dataset REW is curated to facilitate the training specifically for the task of visual entity recognition (VER).

**Strengths:**

1. With the LLM augmented dataset, this work achieves superior performance over previous PALI and other generative models on the task of Oven-Wiki. Additional experiments on fine-grained image classification also validate the effectiveness of the collected dataset.

2. Filtering an entity-centric dataset within three steps: correction, entity rationale generation, and QA generation, could be of great help for future works that focus on synthetic datasets towards entity-specific domains. While rationale and QAs do help, the most significant performance gain for entity split arises from the correction and verification parts, according to Table 4.

3. Extensive ablations studies confirm that additional rationales and QA pairs help with multi-tasking training on VER.

**Weaknesses:**

1. In L241, authors state that the trained model is better than MLLM itself which is used to produce filtered datasets. However, I did not see the results for Gemini Pro (in L226 they mention that) in Table 1.

2. Lack of references: In L142 and L170, authors state that constrained decoding is used to guarantee the successful grounding of entity rationale generation. Works like GMEL [1] and AutoVER [2] should be the first to introduce such techniques into entity generation in multimodal contexts.

3. Minor issue: This work mainly proposes a data-centric solution to GER [3], and I assume the base model architecture of GER and this work stays the same with a 0.4B decoder-only model and a CLIP visual encoder. In Table 1, why is GiT-Large with Entity-WebLI training getting a better performance compared to GER-ALD with the same dataset? Is it because of the newly added two supervisions L_Rationale and L_QA?


[1] Shi S, Xu Z, Hu B, et al. Generative multimodal entity linking[J]. arXiv preprint arXiv:2306.12725, 2023.

[2] Xiao Z, Gong M, Cascante-Bonilla P, et al. Grounding Language Models for Visual Entity Recognition[J]. arXiv preprint arXiv:2402.18695, 2024.

[3] Caron, Mathilde, et al. "A Generative Approach for Wikipedia-Scale Visual Entity Recognition." Proceedings of the IEEE/CVF Conference on Computer Vision and Pattern Recognition. 2024.

**Questions:**

See weaknesses.

**Limitations:**

See weaknesses.

---

> ### Author Rebuttal · Authors · 2024-08-06
>
> **In L241, authors state that the trained model is better than MLLM itself which is used to produce filtered datasets. However, I did not see the results for Gemini Pro (in L226 they mention that) in Table 1.**
>
> This result is shown in the inline text in Section 4.2: “Finally, we report the zero-shot performance of the multimodal LLM on OVEN: it reaches 13.3 HM top-1 in the entity split and 29.5 HM top-1 in the query split”. Note that this is the zero-shot Gemini Pro model, which does not always generate entities that are sufficiently fine-grained. On the other hand, our fine-tuned GiT-Large model achieves 29.6 top-1 in entity split and 30.9 top-1 in the query split. We will make this more clear in the camera ready.
>
> **Lack of references: In L142 and L170, authors state that constrained decoding is used to guarantee the successful grounding of entity rationale generation. Works like GMEL [1] and AutoVER [2] should be the first to introduce such techniques into entity generation in multimodal contexts.**
>
> We thank the reviewer for bringing these papers to our attention, we will definitely discuss them in the camera ready. Note, however, that they differ significantly, as they use retrieval to augment their models which is complementary to our approach.
>
> **Minor issue: This work mainly proposes a data-centric solution to GER [3], and I assume the base model architecture of GER and this work stays the same with a 0.4B decoder-only model and a CLIP visual encoder. In Table 1, why is GiT-Large with Entity-WebLI training getting a better performance compared to GER-ALD with the same dataset?**
>
> The GER paper [6] does not utilize constrained decoding. We opted to apply constrained decoding to both the "GiT-Large with Entity-WebLI" baseline and our model ("GiT-Large with REW-47M") as it yielded a significant performance boost (+5 points) for these captioning models, while it doesn't change the performance of the GER models.

---

> ### Comment · Reviewer_LX9i · 2024-08-07
>
> The author's rebuttal was received. Thanks for clarifying the issues.
>
> Considering all the merits and weaknesses, I decided to update my score (6 -> 7) as it seems to be a fair evaluation of this work.

---

### Official Review · Reviewer_MtgR · 2024-07-12

**Soundness:** 3
**Presentation:** 3
**Contribution:** 3
**Rating:** 6
**Confidence:** 5

**Summary:**

The paper deals with web-scale visual entity recognition, which consists of math a question(text)-image query to one of the 6M entities (wikipedia page title) of a base of reference. In the vein of previous works, the task is addressed with a generative text-to-image model, the challenge lying in building the training dataset of this model and the method to learn it (task/losses considered). The approach is based on the recently published dataset Entity-Web-LI which associates an entity (name) to the images of an image-caption dataset. The novelty lies in an additional module of curation that relies on a multimodal LLM (Gemini), which asserts the relevance of the image retrieved in the image-caption external dataset. The multimodal LLM is also asked to provide explanations on its curation. With the novel training dataset, a Git-Large model is trained and evaluated in the context of visual entity recognition (OVEN benchmark) and zero-shot fine-grained classification (5 benchmarks) with better performances than the recent paper the work is built. The work is completed by several experiments of analysis and ablation.

**Strengths:**

* the proposed approach is built upon a method [6] recently published at CVPR 2024, that is after the NeurIPS 2024 submission deadline. All the experimental evaluations are compared to this paper with a similar model (GIT-Large). They also report the results that [6] obtained with another model (GER-ALD) that has a similar complexity (400M parameters) but obtained better results that Git-Large on the task of large-scale entity recognition. In all cases, the proposed model has better performance.

* the approach is evaluated on both the task of large-scale entity recognition (OVEN query and OVEN entity) and on five benchmarks of fine-grained classification in a zero-shot setting, with better performances than [6] in all cases, according to all metrics. It is worth noting that the experimental results on (zero-shot) fine-grained classification are new ([6] did not report such results) and that much more metrics/settings are reported in this paper than in [6] on OVEN. The authors also report experiments of visual matching with CLIP and Dinov2 on six benchmarks, showing that the proposed dataset allows a significant boost in performance.

* the analyses and ablation in Section 4.3 are quite detailed and give a fair view of the contribution of each method component as well as its behavior in "degraded mode". In particular, using their smaller dataset, the authors report results using LAION

**Weaknesses:**

* It is regrettable to rely on a private model (Gemini-pro) to build the main contribution of the paper, which is the training dataset. There's no guarantee that this model will be stable over time, such that the proposed method (to build the dataset) may not be reproducible in a couple of months. This issue would nevertheless be partially mitigated if the dataset itself is released, although it still limits the interest of the method itself (e.g maybe another prompt would be required to get comparable results, or another multi-task learning to train the T2I model...). Surprisingly, this aspect is not addressed in the "Limitations" of Section 5.

* The authors argue (lines 239-243) that "the zero-shot performance of the multimodal LLM on OVEN" is much lower than those obtained when it is used alone thus it "suggest[s] that we are not merely distilling from the considered multimodal LLM". However, it is hard to believe that the gain in performances over [6] is not essentially due to the use of an additive (large, complex and trained on many data) LLM. Beyond the fact the namely LLM is opaque (see above) the contribution seems to mainly consist of a RAG-like approach to filter and correct the entities associated with the images by [6], relying on a black box which makes the process difficult to actually understand. So yes, "it works", but the scientific contribution nevertheless seems limited in that sense.


**minor**
* line 543 "if you answer" --> "if your answer"
* in Appendix A.5, the authors report that the mapping used for fine-grained classification was that proposed by OVEN "then improved it through a careful manual review". For the sake of reproducibility, it should be clear that this mapping will be released to the community.

**Questions:**

* Can we have an idea of the additional computational complexity (and resource usage) due to Gemini?

* In the same vein, is it possible to estimate the performance boost/drop with another external "multimodal LLM"?

* The authors report that the code is not released (Checklist "Experimental Result Reproducibility") but do they at least plan to release the new "mapping" manually curated and used for fine-grained classification?

**Limitations:**

Lines 324-332 are specifically dedicated to the limitations of the approach. It highlights the dependence of the approach on the availability of relevant external knowledge (that is Wikipedia in the context of the paper) as well as the fact that the proposed approach is expensive in terms of computations. One can regret that the message is vague and generic, without specific insight into the actual proposed method, not to mention possible quantitative hints on the actual computational time (of complexity in terms of memory). It refers to Appendix A.2 but that last focuses again on the performance of the model. However, Appendix A.3.3 reports the usage of 256 TPUv3 during 15+44 hours for the models used in the paper. This only concerns the training of the GitLarge models, but an estimation of the additive resources used to build the dataset (that is the main novelty of the paper) with Gemini would also be relevant.

---

> ### Author Rebuttal · Authors · 2024-08-06
>
> **It is regrettable to rely on a private model (Gemini-pro) to build the main contribution of the paper, which is the training dataset. There's no guarantee that this model will be stable over time, such that the proposed method (to build the dataset) may not be reproducible in a couple of months. This issue would nevertheless be partially mitigated if the dataset itself is released, although it still limits the interest of the method itself (e.g maybe another prompt would be required to get comparable results, or another multi-task learning to train the T2I model...).**
>
> We agree and appreciate the reviewer's valuable feedback regarding open-source accessibility. To address this, we adapted our method to leverage the open-source Gemma 27B model, renowned for its strong reasoning capabilities and long context support. Please see the “General response” of this rebuttal for the results with PaliGemma and Gemma models. We will release the implementation of this variant.
>
>
>
> **The authors argue (lines 239-243) that "the zero-shot performance of the multimodal LLM on OVEN" is much lower than those obtained when it is used alone thus it "suggest[s] that we are not merely distilling from the considered multimodal LLM". However, it is hard to believe that the gain in performances over [6] is not essentially due to the use of an additive (large, complex and trained on many data) LLM. Beyond the fact the namely LLM is opaque (see above) the contribution seems to mainly consist of a RAG-like approach to filter and correct the entities associated with the images by [6], relying on a black box which makes the process difficult to actually understand. So yes, "it works", but the scientific contribution nevertheless seems limited in that sense.**
>
> We would like to clarify the following points:
> 1. We compare the results of Gemini to our model (13.3 versus 29.6 top-1 HM in the entity split). The gain is due to the fact that Gemini is a general purpose model, whereas our model is trained on a Gemini curated dataset for the specific task of entity recognition. So Gemini doesn’t work out of the box, but is able to curate the labels when used properly (more on this in point 3 below).
> 2. We train the same GiT-Large model on two versions of the WebLI datasets, i.e curated and uncurated. We can see that the curation improves the results significantly (20.1 versus 29.6).
> 3. Our scientific contributions in this work is about how to use LLMs for the curating data for web-scale entity-recognition. As evidenced by the results presented in Table 4, our findings highlight two key contributions for the community:
>     * LLMs excel in verification over direct prediction:  We show that a naive application of LLMs for direct prediction (Table 4, Row 1) can actually hinder performance compared to [6]. This is because they tend to generate generic entities, rather than very fine-grained ones required for this task. However, utilizing LLMs for verification purposes (Table 4, Row 3) leads to a significant performance boost.
>     * External knowledge enhances LLM reasoning for fine-grained entities:  We demonstrate that augmenting LLMs with external knowledge sources like Wikipedia (Table 4, Row 4) further improves their ability to reason about and identify fine-grained entities.
>
> To also emphasize the generalizability of our approach beyond a specific LLM, we've included additional experiments with an open-sourced model in the “General response” of this rebuttal. This highlights the adaptability of our methodology across different LLM architectures, even when their internal workings are opaque.
>
> **Is it possible to estimate the performance boost/drop with another external "multimodal LLM"?**
>
> We report the accuracy of our method with other open source models (PaliGemma + Gemma) in Table R3 of the “General response”. Even though there is a small drop in performance when using open source models, our method still outperforms the prior work.
>
> **The authors report that the code is not released (Checklist "Experimental Result Reproducibility") but do they at least plan to release the new "mapping" manually curated and used for fine-grained classification?**
>
> Yes, the mapping in the appendix section A.5 of the paper will be released.

---

### Official Review · Reviewer_Poha · 2024-07-12

**Soundness:** 2
**Presentation:** 2
**Contribution:** 2
**Rating:** 6
**Confidence:** 1

**Summary:**

The paper presents a method to curate datasets for visual entity recognition tasks. They rely on a multimodal LLM (Gemini Pro), which employs metadata information about the image (the caption) and the content of the Wikipedia page to improve the quality of the Entity Web-Li dataset. They further enrich the resource with the rationale regarding the relation between the image and the entity and several question-answer pairs about diverse range of entities appearing in the image.

**Strengths:**

The paper presents a methodology for improving the quality of the Entity Web-Li dataset and show its usefulness in a series of downstream experiments and ablation studies (on the OVEN benchmark and a series of finegrained datasets). The experiments are carried out thoroughly and well described.

**Weaknesses:**

The main contribution of the paper is the creation of an enriched version of the Entity Web-Li dataset, named REW, which however is not released to the public together with the paper (this is mentioned in the Checklist). I find this a major weakness of this project, given the complexity of the conducted work (as clearly described in the paper and the appendix) and would strongly encourage the authors to reconsider this decision.

**Questions:**

Could you clarify how the avg. relative improvement in table 3 is computed?

**Limitations:**

The authors do not clarify why they are not releasing the updated version of the dataset. if this is done for safety reasons, it should be clarified as part of the limitations.

---

> ### Author Rebuttal · Authors · 2024-08-06
>
> **The main contribution of the paper is the creation of an enriched version of the Entity Web-Li dataset, named REW, which however is not released to the public together with the paper (this is mentioned in the Checklist). I find this a major weakness of this project, given the complexity of the conducted work (as clearly described in the paper and the appendix) and would strongly encourage the authors to reconsider this decision.**
>
> We thank the reviewer for their comment. We would like to point out that we also conduct experiments with publicly available LAION dataset (Table 5). Our intention is to release the annotations for the LAION dataset. Furthermore, our results are reproducible with the public Gemini-API, and now also with the open source models described in the “General response” of the rebuttal.
>
>
> **Could you clarify how the avg. relative improvement in table 3 is computed?**
>
> The relative improvement for each dataset is (new_value - old_value) / old_value.
> For each row, we compute the relative improvement separately for each of the six datasets in the table and then compute their average.

---

> > ### Comment · Reviewer_Poha · 2024-08-08
> >
> > Thank you for your reply. Please do release together with the paper the curated annotations for LAION. Regarding reproducibility more generally, I would not underestimate the complexities of reproducing results that are obtained from a proprietary model available only through an API access. For future research, I would highly encourage the authors to also include an available open source model with visual input (e.g. LlaVa).

---

> ### Author Response · Authors · 2024-08-08
> **Response to the Official Comment by Reviewer Poha**
>
> We thank the reviewer for starting the discussion.
>
> We would like to emphasize that we have already demonstrated the performance of our method using open-source models (PaliGemma+Gemma) in Tables R1, R2, and R3 of the "General Response".
>
> Open-source VLMs like PaliGemma and LlaVa do not support long contexts. To overcome this complexity, as the reviewer rightly points out, we employ a two-stage approach. First, we utilize PaliGemma (similar to LlaVa) to process the visual input and generate a detailed caption. Then, we feed the output from PaliGemma, along with with our prompt (Section A.3.1), to a more powerful LLM, Gemma 27B, chosen for its longer context support and stronger reasoning abilities. This two-stage approach allows us to effectively use our methodology with open-source models.
>
> We can see in Table R1 that in all cases our method gives substantial improvements over the prior work GER-ALD[6] and GIT-Large trained on Webli-Entity[6]. We will include the Gemma results with the full 47M dataset in the camera ready.
>
> We hope that this experiment and explanation addresses the reviewer's concerns. We are happy to elaborate further or discuss any other questions the reviewer might have.

---

> > ### Comment · Reviewer_Poha · 2024-08-08
> >
> > Thank you for the detailed answers - yes please do include the Gemma results and publish the curated annotations for LAION. I'm happy to increase my score to 6

---

### Author Rebuttal · Authors · 2024-08-06

We thank the reviewers for their constructive comments. The reviewers especially appreciate the “thorough experiments and ablation studies”, that our work “could be of great help for future works that focus on synthetic datasets towards entity-specific domains”, and better results compared to the previous methods.

We would like to address a few comments regarding the reproducibility of our approach. Firstly, we would like to point out that the results on the public LAION dataset are included in Table 5 (Right). We plan to release the curated annotations for this dataset.

Secondly, we have run an additional set of experiments for the rebuttal with the open source PaliGemma and Gemma 27B models and the results are inline with the results when using the Gemini-Pro model. Since Gemma lacks visual input processing, we replaced direct image input with automatically generated captions. Specifically, we employed the open-source PaliGemma model to generate descriptive captions for each image using the prompt: "Describe the visual attributes of this image."  We then integrated these captions into the existing prompts outlined in Section A.3.1 by prepending the text: "Here are the visual attributes of the image: {paligemma_output}." We plan to release the code for this implementation.

We evaluated this approach on the 5M subset of WebLI and LAION as used in our ablation studies (Section 4.3). We compare the Gemma and Gemini-Pro variants of our method in Table R1 with the SOTA methods trained on the 5M subset of Entity-WebLI. We can see that in all cases our method gives substantial improvements over the prior work GER-ALD[6] and GIT-Large trained on Webli-Entity[6]. We will include the Gemma results with the full 47M dataset in the camera ready.

More detailed analyses are shown in Tables R2 and R3 of the rebuttal. Numbers in bold are the numbers from with Gemma, compared to unbolded numbers from the Table 5 (right) of the paper. We can observe that different losses improve the performance in a similar way. Biggest difference of performance between Gemini-Pro and Gemma variants comes from the QA loss. While Gemini-Pro has access to input images when generating QA pairs, Gemma generates QA pairs based on the PaliGemma caption (and rationale). This limits the variety of the generated QA pairs, resulting in a lower final accuracy.

```
```
>**Table R1: Comparison to the prior work when using WebLI-5M and LAION-5M as the pretraining data.**
|                      | Pre-training Dataset | Entity Split HM | Query Split HM |
| :-------------------- | :-------------------- | :-------------: | :-------------: |
| **SOTA METHODS**         |                      |                 |                 |
| GER-ALD [6]           | WebLI-5M             |       10.2      |        -        |
| GiT-Large (Captioning) | WebLI-5M             |        9.1      |        5.6      |
| **OURS**             |                      |                 |                 |
| REW (Gemma)          | LAION-5M             |       **11.6**      |      **23.4**      |
| REW (Gemini-Pro)     | LAION-5M             |       13.4      |       28.2      |
| REW (Gemma)          | WebLI-5M             |       **14.2**      |       **24.3**      |
| REW (Gemini-Pro)     | WebLI-5M             |      16.0     |      28.2    |
```
```
> **Table R2: Accuracy of generated rationales and QAs in LAION-5M.**
| Entity Loss | Rationale Loss | QA Loss | Gemini-Pro Entity Split | Gemini-Pro Query Split | Gemma Entity Split | Gemma Query Split |
|---|---|---|---|---|---|---|
| YES |  |  | 10.7 | 5.6 | **9.5** | **9.4** |
| YES | YES |  | 11.4| 6.9 | **10.6** | **9.7** |
| YES | YES | YES | 13.4 | 28.2 | **11.6** | **23.4** |
```
```
> **Table R3: Accuracy of generated rationales and QAs in WebLI-5M.**
| Entity Loss | Rationale Loss | QA Loss | Gemini-Pro Entity Split | Gemini-Pro Query Split | Gemma Entity Split | Gemma Query Split |
|---|---|---|---|---|---|---|
| YES |  |  | 14.1 | 5.4 | **11.9** | **5.8** |
| YES | YES |  | 14.6 | 6.7 | **13.3** | **6.3** |
| YES | YES | YES | 16.0 | 28.2 | **14.2** | **24.3** |
```
```
We address the reviewers’ comments in more detail in the corresponding sections.

---

### Decision · Program_Chairs · 2024-09-25

**Decision:**

Accept (poster)

**Comment:**

The paper originally mixed reviews, although they were definitely leaning towards acceptance. The authors provided a detailed rebuttal and the paper has been discussed by the reviewers. The reviewers raised some concerns about missing experiments and some minor commment about presentation. The reviewers also asked for clarification about the collected dataset, and criticized the decision of relying on a private model (Gemini-pro) to build the dataset. The authors included in their responses new results and additional experiments with an open model; the reviewers where satisfied by that and, more in general, by the authors' rebuttal and they all maintained or increased their scores. Therefore there is a general consensus towards accepting the paper. The AC believes this is a solid submission and the authors are encouraged to carefully follow and implement all the reviewers' suggestion in the final revision of the paper.